# Exact Functional ANOVA Decomposition for Categorical Inputs Models

**Baptiste Ferrere** [* 1 2]  **Nicolas Bousquet** [† 1 3]  **Fabrice Gamboa** [† 2 4 5]  **Jean-Michel Loubes** [† 2 6 4]  **Joseph Muré** [† 1]

## Abstract

Functional ANOVA offers a principled framework for interpretability by decomposing a model's prediction into main effects and higher-order interactions. For independent features, this decomposition is well-defined, strongly linked with SHAP values, and serves as a cornerstone of additive explainability. However, the lack of an explicit closed-form expression for general dependent distributions has forced practitioners to rely on costly sampling-based approximations. We completely resolve this limitation for categorical inputs. By bridging functional analysis with the extension of discrete Fourier analysis, we derive a closed-form decomposition without any assumption. Our formulation is computationally very efficient. It seamlessly recovers the classical independent case and extends to arbitrary dependence structures, including distributions with non-rectangular support. Furthermore, leveraging the intrinsic link between SHAP and ANOVA under independence, our framework yields a natural generalization of SHAP values for the general categorical setting. We provide a basic Python implementation of our method.

## 1. Introduction

Model explanation is essential to understand, compare, and validate the mechanisms underlying predictive performance. It can be local or global and may aim either at importance attribution, such as Shapley Additive exPlanations (SHAP values, Lundberg & Lee (2017)), or at the identification of the effects and interactions governing model behavior (Il Idrissi, 2024). It traditionally faces three challenges: ensuring a theoretically grounded explanation, controlling the complexity of interactions, and guaranteeing alignment

between model behavior, data structure, and the produced explanation. Building on these observations, several principled frameworks to characterize latent factors and interactions have been proposed (Khemakhem et al., 2020; Liang et al., 2023). In the post-processing context of black-box models (*post-hoc explainability*), functional ANOVA offers such decomposition framework with broad relevance to these challenges.

More specifically, functional ANOVA was initially introduced by Hoeffding (1948) for independent random variables then generalized by Stone (1994) and later by Hooker (2007) for dependent random variables, providing a global additive decomposition that non-redundantly separates effects and interactions. Finally, Chastaing et al. (2012) then Il Idrissi et al. (2025) worked on the existence of this decomposition for unbounded support while preserving its uniqueness.

Recent work aims to make such additive explanations computationally tractable at scale in specific settings. In particular, for binary inputs, the sparse Fourier representation proposed by Gorji et al. (2024) based on the Boolean analysis (O'Donnell, 2014) enables fast and *exact* SHAP values computation. Their `FourierSHAP` values are however restricted to the *Interventional* (Janzing et al., 2020; Van den Broeck et al., 2022) or *Baseline* (Sundararajan & Najmi, 2020) Shapley values. In the context of functional ANOVA decomposition, Boolean analysis cannot be applied directly for two major reasons. First, computing the Fourier decomposition of pseudo-Boolean functions is equivalent to performing an ANOVA decomposition on functions with *i.i.d.* Bernoulli inputs of parameter $1/2$, a condition that is rarely met in practice. Second, in the categorical case, any one-hot encoding of complex categorical inputs leads to the study of fictitious interactions between the *new* binary variables, due to the corresponding encoding. Consequently, one cannot perform *brute-force* boolean Fourier analysis to perform functional ANOVA decomposition.

In a more general input setting, but restricted to tree ensembles, Lengerich et al. (2020) introduced the `Purifying` algorithm to estimate the functional ANOVA decomposition, but this method requires the density of the data. Later, Bénard (2025) proposed the `TreeHFD` algorithm which directly estimates the decomposition from a data sample. However, this approach is computationally limited to shal-

---

*Corresponding author. †Equal supervising. [1]EDF R&D, SIN-CLAIR Lab [2]Université de Toulouse [3]Sorbonne Université [4]ANITI [5]Facultad de Ingeníera Universidad de Medellín [6]INRIA Regalia. Correspondence to: Baptiste Ferrere <baptiste.ferrere@edf.fr>.

*Proceedings of the 43rd International Conference on Machine Learning*, Seoul, South Korea. PMLR 306, 2026. Copyright 2026 by the author(s).

low trees and assumes non-empty leaves, as is standard for this class of models (Kairgeldin & Carreira-Perpiñán, 2024), which precludes accounting for sparsity in the data.

To leverage the properties of this generalized Fourier decomposition more broadly in practical applications, a computationally tractable formulation is desirable. This is particularly relevant when inputs are categorical and may be dependent. Categorical variables are ubiquitous in tabular data and require consequential representation choices (Matteucci et al., 2023), which typically induce high cardinality and out-of-vocabulary modalities (Cheng et al., 2025). Moreover, studying categorical-input settings is a natural stepping stone toward continuous inputs in modern tabular models, since competitive approaches unify feature processing through an embedding/tokenization stage for both categorical and numerical attributes, and show that even continuous variables benefit from vector encodings before the backbone (Gorishniy et al., 2021).

This setting is precisely the focus of this work. We explicitly construct this generalized decomposition in a computationally tractable manner, and show that it can handle high-dimensional inputs without sacrificing the need to account for dependencies among them. We provide theoretical guarantees and extensive experiments. Our contributions are summarized below.

**Our contributions.** We introduce a closed-form formulation for the Generalized Functional ANOVA on categorical domains. To the best of our knowledge, this constitutes a novel theoretical advance: it provides an exact additive decomposition, valid for any arbitrary dependence structures and sparse empirical supports, where standard methods typically fail or rely on approximations. We demonstrate that this general framework naturally recovers established results, such as orthogonal ANOVA and SHAP values, in the independent setting. Furthermore, we exploit the vectorized structure of our framework and empirical sparsity of tabular categorical data to derive efficient model explanation. A visual example is provided on Fig 1. This framework paves the way for a unified, theoretically-grounded and computationally tractable local and global explainability.

## 2. Background

**Notations.** Let $d$ be a positive integer and denote $[d] \coloneqq \{1, \ldots, d\}$. We consider a random input vector $\mathbf{X} \coloneqq (\mathbf{X}_1, \ldots, \mathbf{X}_d)$ taking values in a support $\mathcal{X} \subseteq \mathbb{R}^d$, with probability density function $p$ defined with respect to a measure $\nu$. For any subset of indices $A \subseteq [d]$, let $\mathbf{X}_A \coloneqq (\mathbf{X}_i)_{i \in A}$ be the corresponding subvector. We define $L^2(p)$ as the Hilbert space of square-integrable measurable functions of $\mathbf{X}$, equipped with the standard inner product $\langle f, g \rangle \coloneqq \mathbb{E}[f(\mathbf{X})g(\mathbf{X})] = \int_{\mathcal{X}} f(\mathbf{x})g(\mathbf{x})p(\mathbf{x})d\nu(\mathbf{x})$.

Following the foundational work of Hoeffding (1948) and Stone (1994), we focus on the Functional ANOVA decomposition, generalized later by Hooker (2007) as a variational problem, properly defined below.

**Generalized Functional ANOVA (Hooker, 2007).** Let $f \in L^2(p)$. We say that $f$ satisfies a generalized functional ANOVA decomposition if we can jointly define a collection of functions $\{f_A\}_{A \subseteq [d]}$ such that

$$f(\mathbf{X}) = \sum_{A \subseteq [d]} f_A(\mathbf{X}_A), \tag{1}$$

subject to the *hierarchical orthogonality condition*

$$\forall B \subsetneq A, \ \forall g \in L_B^2, \quad \langle\, f_A(\mathbf{X}_A)\,,\, g(\mathbf{X}_B)\,\rangle = 0, \tag{2}$$

where $L_B^2$ denotes the subspace of functions in $L^2(p)$ depending only on the variables indexed by $B$.

The hierarchical orthogonality condition (2) is paramount: it ensures that any information added by a higher-order set $A$ is strictly new (orthogonal) to the information already contained in any proper subset $B \subsetneq A$, thereby leading to a decay of information per set cardinality.

Under suitable conditions for its existence, obtaining the decomposition (1) while satisfying the hierarchical orthogonality conditions (2) is a challenging task. While no closed-form formula exists for the general case, the independent setting admits an explicit and unique solution. In this context, the decomposition components are mutually orthogonal and are retrieved via the Möbius transform (Rota, 1964) of the conditional expectations

$$f_A(\mathbf{X}_A) = \sum_{B \subseteq A} (-1)^{|A|-|B|} \mathbb{E}\left[f(\mathbf{X}) \mid \mathbf{X}_B\right]. \tag{3}$$

**Shapley values (Owen, 2014; Owen & Prieur, 2017).** For interpretability tasks and alternatively to the standard formulation, Shapley values (Shapley, 1953) can be defined through the lens of *dividends* (Harsanyi, 1963). Let $w$ be a set function such that the value of any coalition $S \subseteq [d]$ decomposes as $v(S) = \sum_{A \subseteq S} w(A)$. The corresponding Shapley value $\mathrm{shap}_i$ for player $i \in [d]$ is then uniquely determined by distributing these dividends equally among the participants

$$\mathrm{shap}_i \coloneqq \sum_{A \subseteq [d]\,:\,A \ni i} \frac{w(A)}{|A|}. \tag{4}$$

Leveraging this perspective, one can define Shapley values for *local feature attribution*. For a specific query $\mathbf{x}$, the components $f_A(\mathbf{x}_A)$ from the Generalized Functional ANOVA act as the Harsanyi dividends of the model. The

Original Input

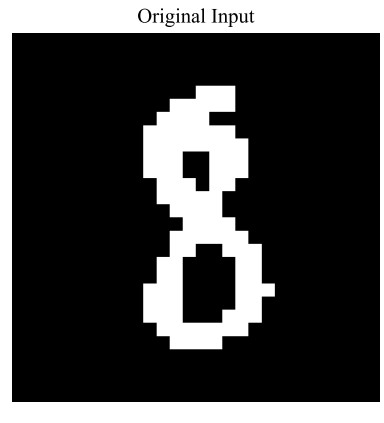

Feature Contribution

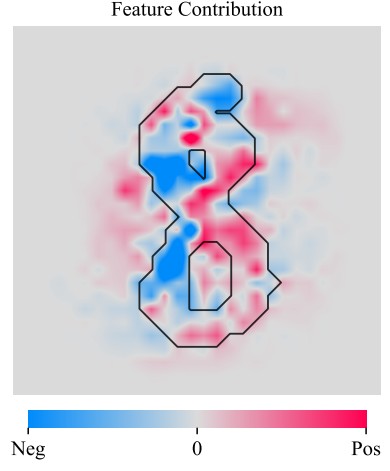

Neg      0      Pos

Absolute Importance

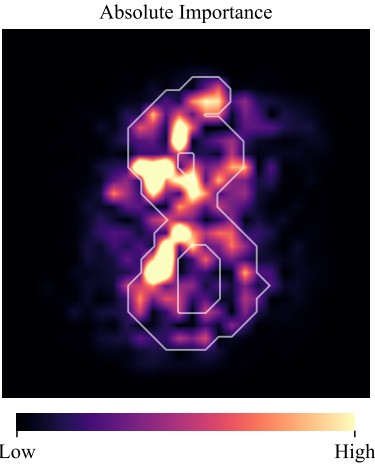

Low      High

*Figure 1.* **ANOVA-Based Shapley Values on the Binarized MNIST Dataset.** We apply our framework on a MLP trained on binarized MNIST, encoded as a tabular dataset of shape $(60\,000, 784)$. The attribution targets the predicted probability of the specific class '3', defined as $f(\mathbf{x}) := \mathbb{P}(MLP(\mathbf{x}) = 3)$. **(Left)** The original input sample (digit '8'). **(Middle)** Signed local attributions: red pixels increase the probability of the target class '3', while blue pixels decrease it. **(Right)** Absolute attribution magnitudes. We observe a behavior consistent with the nature of the digits. The pixels on the right side, which overlap with the shape of a '3', positively contribute to the target probability (red). Conversely, the pixels on the left side close the loops, acting as the distinguishing features between an '8' and a '3'; these pixels correctly penalize the target probability (blue), allowing the model to rule out class '3'.

corresponding Shapley values $\text{shap}_i(\mathbf{x})$ are given by

$$\text{shap}_i(\mathbf{x}) := \sum_{A \subseteq [d]\,:\,A \ni i} \frac{f_A(\mathbf{x}_A)}{|A|}. \quad (5)$$

This ANOVA-based formulation then can be seen as a generalization of SHAP values (Lundberg & Lee, 2017) in the particular case where the components of $\mathbf{X}$ are mutually independent.

## 3. Main Theoretical Contribution

In this paper, we specialize the previous framework to the case where inputs $\mathbf{X}$ are **categorical**. In this setting, the measure $\nu$ is the counting measure. The support $\mathcal{X}$ is finite and is contained in the *full hypergrid* $E$ defined as

$$E := \prod_{i=1}^{d} \{0, \ldots, N_i - 1\}, \quad (6)$$

where $N_i$ denotes the number of categories for variable $X_i$. Consequently, $p$ becomes a probability mass function (pmf). Note that in the general case (e.g., dependent variables), the support $\mathcal{X}$ may be a strict subset of the hypergrid ($\mathcal{X} \subsetneq E$).

For any subset $A \subseteq [d]$, we define the *truncated grid* $E_{A^-} := \prod_{i \in A} \{0, \ldots, N_i - 2\}$, corresponding to configurations where the last category is excluded. The total cardinality of the hypergrid admits the following decomposition

based on the *inclusion-exclusion principle* (Rota, 1964)

$$|E| = \prod_{i=1}^{d} N_i = \sum_{A \subseteq [d]} \prod_{i \in A} (N_i - 1) = \sum_{A \subseteq [d]} |E_{A^-}|. \quad (7)$$

This motivates the definition of the index space

$$\mathcal{I} := \{(A, \mathbf{z}) \,:\, A \subseteq [d], \mathbf{z} \in E_{A^-}\},$$

which is in bijection with $E$. Note that $\mathbf{z}$ depends on the set $A$ since it is valued in $E_{A^-}$. Finally, since $E$ is finite, the corresponding Hilbert space denoted $L^2$ encompasses all finite real-valued functions defined on the support $\mathcal{X}$.

### 3.1. Closed-Form Representation

We introduce in next definition an extension of the Walsh–Hadamard basis, also known as the family of parity functions, that plays a key role in the Fourier analysis of Boolean functions (O'Donnell, 2014). This extension has the nice property of satisfying the condition (2).

**Definition 3.1** (Inverse likelihood mechanism). Let $\mathcal{F} := \{\phi_A^{(\mathbf{z})}, (A, \mathbf{z}) \in \mathcal{I}\}$ be the collection of functions $\phi_A^{(\mathbf{z})}$ defined for any $\mathbf{x} \in \mathcal{X}$ by

$$\phi_A^{(\mathbf{z})}(\mathbf{x}) := \frac{\prod_{i \in A} \left( \mathbf{1}\{\mathbf{x}_i = \mathbf{z}_i\} - \mathbf{1}\{\mathbf{x}_i = N_i - 1\} \right)}{p_A(\mathbf{x}_A)}$$

By convention, $E_{\emptyset^-}$ is a singleton and the only element $\mathbf{z}_\emptyset \in E_{\emptyset^-}$ satisfies $\phi_\emptyset^{(\mathbf{z}_\emptyset)} = 1$.

Note that the denominator is strictly non-zero, provided that $\mathbf{x}$ lies within the support $\mathcal{X}$ of the random variable $\mathbf{X}$. Furthermore, this function can be interpreted as a *signed inverse likelihood*. The numerator acts as a sign function, taking values in $\{-1, 0, 1\}$, while the denominator corresponds to the likelihood of $\mathbf{X}_A$.

---

**Theorem 3.2.** *Any function $f \in L^2$ admits a Fourier expansion of the form*

$$f(\mathbf{X}) = \sum_{(A,\mathbf{z}) \in \mathcal{I}} c_A^{(\mathbf{z})}(f) \cdot \phi_A^{(\mathbf{z})}(\mathbf{X}) \qquad (8)$$

*where the set of coefficients $\{c_A^{(\mathbf{z})}(f)\}$ solves the linear problem formulated in Section 3.3. Moreover, the collection of functions defined by*

$$f_A := \sum_{\mathbf{z} \in E_{A^-}} c_A^{(\mathbf{z})}(f) \cdot \phi_A^{(\mathbf{z})}, \qquad (9)$$

*satisfies the functional ANOVA formulation (1) and the hierarchical orthogonality condition (2).*

---

*Remark* 3.3. Obviously, for all $A \neq \emptyset$, $f_A$ are centered and $c_\emptyset^{(\mathbf{z}_\emptyset)}(f) = \mathbb{E}\left[f(\mathbf{X})\right]$

The preceding theorem establishes that the family $\mathcal{F}$ constitutes a *spanning set* for the Generalized Functional ANOVA decomposition. While the proof leverages linear algebraic arguments to show that $\mathcal{F}$ generates the entire $L^2$ space, it does not imply linear independence of the basis elements. Consequently, without further assumptions on the support $\mathcal{X}$, the uniqueness of this expansion is not guaranteed in general. In practice, this is not a major issue as we prioritize the *simplest possible decomposition*, favoring representations driven by low-order interactions. Empirically, the decomposition is sparse and *main effects* and *pair effects* are often sufficient to effectively reconstruct $f$.

This representation provides a closed-form of the Generalized Functional ANOVA proposed by Hooker (2007). A key advantage of this framework is its ability to extend the standard ANOVA to *complicated settings*, such as functional dependencies among inputs and so in particular a non rectangular support. Furthermore, while it consistently recovers the standard ANOVA in regular regimes, its algebraic structure is particularly amenable to tractable computation, especially when applied to empirical measures (*e.g.* on standard tabular datasets).

In high-dimensional regimes, our approach guarantees, at a minimum, to recover the unique additive decomposition

$$f = \underbrace{f_\emptyset + f_1 + \cdots + f_d}_{f_{\text{explainable}}} + \underbrace{f_{\{1,\ldots,d\}}}_{f_{\text{residual}}}, \qquad (10)$$

where the *main effects* are computed with high efficiency while strictly satisfying hierarchical orthogonality conditions. This ensures that the information captured by these first-order components is maximized, thereby optimizing global interpretability.

### 3.2. Fourier Representation to Functional ANOVA

We now state the assumption under which the above representation is unique.

**Corollary 3.4.** *Consider the full-support setting, i.e. where $\mathcal{X} = E$. In this case, the full collection $\mathcal{F}$ forms a basis of $L^2$ and the decomposition (8) becomes unique.*

Since the family is generating and its cardinality matches the dimension of the space, then uniqueness of representation follows immediately. While Chastaing et al. (2012) and subsequently Il Idrissi et al. (2025) established the theoretical existence and uniqueness of the Functional ANOVA decomposition under these assumptions, our result drastically improves this work by providing an *explicit construction* of the decomposition basis for categorical data.

**Corollary 3.5.** *Assume that the input variables are mutually independent. Then, the decomposition (8) strictly recovers the formulation given in (3).*

*Remark* 3.6. In the specific case of $\{0, 1\}^d$ endowed with the uniform measure, our proposed basis collapses to the standard parity functions basis $\{\chi_A\}_{A \subseteq [d]}$, explicitly given by

$$\forall \mathbf{x} \in \{0, 1\}^d, \quad \chi_A(\mathbf{x}) := (-1)^{\sum_{i \in A} \mathbf{x}_i}. \qquad (11)$$

Consequently, our framework strictly recovers the corresponding Fourier decomposition (O'Donnell, 2014).

### 3.3. Linear Formulation

In this subsection, we introduce the algebraic objects and matrix notations required to formulate the underlying linear system. Specifically, we establish that the coefficients $c_A^{(\mathbf{z})}(f)$ arise as the solution to this linear system and can be interpreted as generalized discrete Fourier coefficients.

**Definition 3.7.** Consider two subsets $A, B \subseteq [d]$ and vectors $\mathbf{z}^{(A)} \in E_{A^-}$ and $\mathbf{z}^{(B)} \in E_{B^-}$. We define the entry $\gamma_{A,B}\left(\mathbf{z}^{(A)}, \mathbf{z}^{(B)}\right)$ as the following $L^2$ inner product

$$\gamma_{A,B}\left(\mathbf{z}^{(A)}, \mathbf{z}^{(B)}\right) := \left\langle \phi_A^{(\mathbf{z}^{(A)})}, \phi_B^{(\mathbf{z}^{(B)})} \right\rangle. \qquad (12)$$

This induces a Gram matrix $\Gamma$ structured into blocks $\Gamma_{A,B}$, formally defined as

$$\Gamma_{A,B} := \left(\gamma_{A,B}\left(\mathbf{z}^{(A)}, \mathbf{z}^{(B)}\right)\right)_{(\mathbf{z}^{(A)}, \mathbf{z}^{(B)}) \in E_{A^-} \times E_{B^-}}. \qquad (13)$$

Each block $\Gamma_{A,B}$ has dimensions $|E_{A^-}| \times |E_{B^-}|$. Furthermore, the full matrix $\Gamma$ is square, symmetric, and positive semi-definite, with total dimension $|E| \times |E|$.

*Remark* 3.8. In the independent setting, this matrix is block diagonal, *i.e.* $\Gamma = \mathrm{diag}\left(\Gamma_{\emptyset,\emptyset}, \ldots, \Gamma_{[d],[d]}\right)$.

**Definition 3.9.** Consider a subset $A \subseteq [d]$ and a configuration vector $\mathbf{z} \in E_{A^-}$. For any function $f \in L^2$, we define the coefficient $\mu_A^{(\mathbf{z})}(f)$ as the following inner product

$$\mu_A^{(\mathbf{z})}(f) := \left\langle f, \phi_A^{(\mathbf{z})} \right\rangle. \qquad (14)$$

Stacking these terms according to the previously defined block-wise structure as follows

$$\boldsymbol{\mu}_A(f) := \left(\mu_A^{(\mathbf{z})}(f)\right)_{\mathbf{z} \in E_{A^-}}^{\top}, \qquad (15)$$

yields the global mean vector $\boldsymbol{\mu}(f)$ of dimension $|E|$.

**Proposition 3.10.** *The family of real coefficients $c_A^{(\mathbf{z})}(f)$ (see (8)) can be represented as a flattened vector denoted $\boldsymbol{c}(f)$ and the corresponding vector is a solution to the linear system*

$$\Gamma \boldsymbol{c}(f) = \boldsymbol{\mu}(f), \qquad (16)$$

This linear system (symmetric and positive semi-definite) admits at least one solution in general and this solution is unique if and only if $\Gamma$ is invertible, *i.e.*, a full support setting.

## 4. Computing the Decomposition

Let us first assume access to the exact joint probability distribution with full support over the hypergrid. In regimes of moderate cardinality (typically $|E| \lesssim 10^4$), the linear system (16) can be solved using standard computational resources to achieve exact recovery of all the interaction terms. Beyond this threshold, the curse of dimensionality renders the exhaustive computation of the full basis intractable.

More realistically when $d$ increases, let us assume that we only dispose of sparse tabular observations of input configurations and numerical outputs. This can generally occur when a large proportion of input combinations are structurally impossible. This also can occur because the finite nature of a sample means that configurations with a low probability often remain unobserved. In both cases, the mass is concentrated on a subset $\mathcal{X}$ of $E$ whose cardinality is significantly smaller than the full grid $E$, implying an effective dimension lower than the ambient one.

**Scalable decomposition under sparsity.** Let us assume that the input $X$ takes exactly $r$ distinct values, implying $|\mathcal{X}| = r$. In this setting, the dimension of the function space is obviously reduced to $\dim L^2 = r$. Recall that, by Theorem 3.2, the exhaustive collection $\mathcal{F}$ spans the entire space $L^2$. Consequently, fundamental linear algebra principles guarantee the existence of at least one subfamily of exactly

$r$ vectors, extracted from this overcomplete collection, that forms a valid basis. Furthermore, relative to this specific selection of basis vectors, the resulting decomposition is unique. We formalize this result in the next theorem.

**Theorem 4.1.** *Let $r = |\mathcal{X}|$ denote the cardinality of the effective support, there exists at least one subset $\mathcal{S}_r$ of $\mathcal{I}$ satisfying $|\mathcal{S}_r| = r$ such that the corresponding collection*

$$\mathcal{B}_{\mathcal{S}_r} := \left\{ \phi_A^{(\mathbf{z})}, (A, \mathbf{z}) \in \mathcal{S}_r \right\} \qquad (17)$$

*forms a basis of the space $L^2$.*

**Corollary 4.2.** *As a direct consequence of the previous theorem, the function $f$ admits the following decomposition*

$$f(\mathbf{X}) = \sum_{(A,\mathbf{z}) \in \mathcal{S}_r} c_A^{(\mathbf{z})}(f) \cdot \phi_A^{(\mathbf{z})}(\mathbf{X}). \qquad (18)$$

*Moreover, letting $\boldsymbol{c}_r(f)$ denote the vector obtained by flattening the coefficients $\{c_A^{(\mathbf{z})}(f)\}_{(A,\mathbf{z}) \in \mathcal{S}_r}$, we have that $\boldsymbol{c}_r(f)$ is given by solving the restriction of the linear system (16) to the index set $\mathcal{S}_r$.*

*Remark* 4.3. A direct consequence of this theorem is that, for a fixed representation basis, the decomposition of $f$ is unique. Furthermore, standard linear algebra arguments ensure that if $f(\mathbf{X})$ is independent of a subvector $\mathbf{X}_S$, then all coefficients $c_A^{(\mathbf{z})}(f)$ for which $A \cap S \neq \emptyset$ equal 0.

**Identifiability and feature correlation.** It is important to see that the basis support $\mathcal{S}_r$ is not unique. The universe of interactions $\mathcal{I}$ constitutes an *overcomplete dictionary*; while the decomposition coefficients are unique *conditioned* on a fixed support $\mathcal{S}_r$. The selection of the support itself is an ill-posed problem with a huge number of solutions. This is the direct consequence of strong feature correlations within the data. When $r \ll |E|$, the probability mass is concentrated on specific patterns, leaving the vast majority of the hypergrid empty. Mathematically, this implies that for any subset $B$ not included in the selected $\mathcal{S}_r$, any function of $X_B$ collapses into a deterministic linear combination of the basis elements. Consequently, these omitted terms possess no independent degrees of freedom: on the restricted support, they are fully determined by the selected basis configuration and thus do not represent *genuine* interactions.

**Example.** Consider a first toy example with $d = 2$ perfectly correlated Bernoulli variables ($\mathbf{X}_1 = \mathbf{X}_2$ a.s.), where the support is restricted to the diagonal $\mathcal{X} = \{(0, 0), (1, 1)\}$. Here, the dimension collapses to $r = 2$. Selecting a basis becomes an arbitrary *attribution choice*: modeling the signal purely via $X_1$ or purely via $X_2$ yields identical predictions on $\mathcal{X}$, despite offering distinct structural interpretations.

Now consider the modified example where the parameters of the Bernoulli distribution are denoted $q_1$ and $q_2$. We suppose

also that the support satisfies $\mathcal{X} = \{(0,0),(0,1),(1,0)\}$. Here, the effective sample size is $r = 3$, whereas the full combinatorial space size remains $|E| = 4$. Let us denote the outputs of $\phi_A^{(\mathbf{z})}(\mathbf{X})$ as column vectors $u_\emptyset, u_1, u_2, u_{12} \in \mathbb{R}^3$, representing their evaluation on the ordered dataset.

$$\underbrace{\begin{pmatrix} 1 \\ 1 \\ 1 \end{pmatrix}}_{u_\emptyset}, \quad \underbrace{\begin{pmatrix} \frac{1}{1-q_1} \\ \frac{1}{1-q_1} \\ \frac{-1}{q_1} \end{pmatrix}}_{u_1}, \quad \underbrace{\begin{pmatrix} \frac{1}{1-q_2} \\ \frac{-1}{q_2} \\ \frac{1}{1-q_2} \end{pmatrix}}_{u_2}, \quad \underbrace{\begin{pmatrix} \frac{1}{1-q_1-q_2} \\ \frac{-1}{q_2} \\ \frac{-1}{q_1} \end{pmatrix}}_{u_{12}} \quad (19)$$

In this setting, the triplet $(u_\emptyset, u_1, u_2)$ constitutes a natural basis for $\mathbb{R}^3$. We identify this as the *canonical* decomposition—in the sense of simplicity—as it prioritizes main effects and excludes the interaction term, already recovered by the main effects.

**Rank-based construction.** Assume that the joint distribution of $\mathbf{X}$, denoted by $\mathcal{D}$ is given. This implies the knowledge of its support $\mathcal{X}$, its cardinality $r$, and the probability mass associated with each of the $r$ realizations constituting the support. To compute the decomposition basis, we introduce the following constructive rank-based procedure. In practice, however, the true distribution $\mathcal{D}$ is typically unknown. We instead rely on a dataset, represented by a data matrix $\mathbb{X} \in \mathbb{R}^{n \times d}$ containing $n$ realizations of $\mathbf{X}$. Consequently, we rely on the empirical distribution $\widehat{\mathcal{D}}_n$, which induces an empirical support and a corresponding empirical probability mass function. Although the empirical distribution may imperfectly approximate the underlying measure, it is worth noting that the model to be explained was never exposed to the true distribution $\mathcal{D}$. Fundamentally, we are explaining a model trained on the empirical distribution $\widehat{\mathcal{D}}_n$.

---

**Algorithm 1** Greedy Approach

---

**Require:** Distribution $\mathcal{D}$
**Ensure:** Functional basis $\mathcal{B}$ of size $r$
1: Fix an order in $\mathcal{I}$, denoted by $\{I_1, \ldots, I_{|E|}\}$
2: *# Consider canonical set order $\emptyset, \{1\}, \{2\}, \ldots$*
3: Initialize $\ell \leftarrow 1$ and set $(A, \mathbf{z}) \leftarrow (I_\ell[0], I_\ell[1])$
4: $\mathcal{B} \leftarrow \{\phi_A^{(\mathbf{z})}\}$
5: **while** $\text{rank}(\mathcal{B}) < r$ **and** $\ell < |E|$ **do**
6: $\quad \ell \leftarrow \ell + 1$
7: $\quad (A, \mathbf{z}) \leftarrow (I_\ell[0], I_\ell[1])$
8: $\quad$ Let $v_{try} \leftarrow \phi_A^{(\mathbf{z})}$
9: $\quad \mathcal{B}_{try} \leftarrow \mathcal{B} \cup \{v_{try}\}$
10: $\quad$ **if** $\text{rank}(\mathcal{B}_{try}) > \text{rank}(\mathcal{B})$ **then**
11: $\quad\quad \mathcal{B} \leftarrow \mathcal{B}_{try}$
12: $\quad$ **end if**
13: **end while**
14: **return** $\mathcal{B}$

---

**Low-rank approximation.** While Algorithm 1 theoretically proceeds until the full rank $r$ is reached, the dimensionality of the support often renders the exact decomposition computationally prohibitive. In practice, we operate under a budget constraint $r_{\text{low}} < r$. The procedure terminates once the basis cardinality reaches this threshold. Empirically, standard metrics such as $R^2$, MSE, and Relative MSE maintain good to high performance levels despite this truncation. This reduction effectively navigates the accuracy-interpretability trade-off: it distills potentially complex models operating in dense spaces into concise, parsimonious explanations by sacrificing little reconstruction fidelity.

**Computational complexity.** The worst-case complexity of Algorithm 1 is $O(|E| * r^2)$: the procedure iterates over up to $|E|$ candidate basis functions and performs, at each step, a Gram–Schmidt-type rank check whose cost scales as $O(r^2)$ in the current basis size. In practice, two complementary mechanisms make this complexity manageable in high-dimensional settings. First, the low-rank approximation introduced above replaces the full target rank $r$ by a budget $r_{\text{low}} \ll r$, reducing the effective cost to $O(|E| * r_{\text{low}}^2)$. Second, by enumerating candidates in the *canonical* order, Algorithm 1 exhausts low-order interactions before considering higher-order ones; this aligns with the sparsity-of-effects principle, under which the dominant contributions to $f$ typically arise from low-order components, so that the most informative basis elements are selected early.

## 5. Experiments

**Analytical case.** We design a synthetic experiment with $d = 5$ variables to validate our framework. We impose the functional constraints $\mathbf{X}_3 = \mathbf{X}_2$ (perfect dependency) and $\mathbf{X}_5 = 1$ almost surely (constant). For simplicity, we suppose that $\mathbf{X}_1, \mathbf{X}_2, \mathbf{X}_4$ are *i.i.d.* and follow a uniform distribution over $\{0, 1, 2\}$. The target function to be explained is defined as

$$f(\mathbf{X}) := \text{sign}(\mathbf{X}_1 - \mathbf{X}_2 + 0.5 \cdot \mathbf{X}_3). \quad (20)$$

Theoretically, the functional space $L^2$ is strictly spanned by the free variables $(\mathbf{X}_1, \mathbf{X}_2, \mathbf{X}_4)$. Furthermore, due to the redundancy $\mathbf{X}_3 = \mathbf{X}_2$, the target model reduces to a function of $(\mathbf{X}_1, \mathbf{X}_2)$ only.

We evaluate our framework on the support $\mathcal{X}$. As expected, the elements $\phi_A^{(\mathbf{z})}$ selected to construct $L^2$ do not contain $\mathbf{X}_3$ and $\mathbf{X}_5$. Moreover, Table 1 confirms the irrelevance of variable $\mathbf{X}_4$.

**Comparison with KernelSHAP in independent setting.** We use the CAR EVALUATION and NURSERY datasets (details in Table 2). These datasets are characterized by a uniform distribution over the full hypergrid $E$, which the-

*Table 1.* Norms of ANOVA Decomposition elements $f_A$.

| Order | Subset $A$ | Norm $\|f_A\|_2^2$ |
|---|---|---|
| Intercept | $\emptyset$ | 0.111 |
| Main Effects | $\{1\}$ | 0.518 |
| | $\{2\}$ | 0.074 |
| | $\{4\}$ | 0.000 |
| Interactions | $\{1, 2\}$ | 0.074 |
| | $\{1, 4\}$ | 0.000 |
| | $\{2, 4\}$ | 0.000 |
| | $\{1, 2, 4\}$ | 0.000 |

*Table 2.* Uniform categorical datasets.

| DATASET | $d$ | $\{N_1, \ldots, N_d\}$ | $|E|$ |
|---|---|---|---|
| CAR EVALUATION | 6 | $\{4, 4, 4, 3, 3, 3\}$ | 1 728 |
| NURSERY | 8 | $\{3, 5, 4, 4, 3, 2, 3, 3\}$ | 12 960 |

*Table 3.* $100 \times ISE$ on CAR EVALUATION.

| | $\mathbf{X}_1$ | $\mathbf{X}_2$ | $\mathbf{X}_3$ | $\mathbf{X}_4$ | $\mathbf{X}_5$ | $\mathbf{X}_6$ |
|---|---|---|---|---|---|---|
| Class 0 | 2.01 | 1.30 | 0.01 | 0.28 | 0.03 | 0.30 |
| Class 1 | 0.16 | 0.15 | 0.00 | 0.04 | 0.01 | 0.05 |
| Class 2 | 2.89 | 1.46 | 0.02 | 0.71 | 0.05 | 0.71 |
| Class 3 | 0.21 | 0.09 | 0.01 | 0.04 | 0.04 | 0.12 |

*Table 4.* $100 \times ISE$ on NURSERY.

| | $\mathbf{X}_1$ | $\mathbf{X}_2$ | $\mathbf{X}_3$ | $\mathbf{X}_4$ | $\mathbf{X}_5$ | $\mathbf{X}_6$ | $\mathbf{X}_7$ | $\mathbf{X}_8$ |
|---|---|---|---|---|---|---|---|---|
| Class 0 | 0.0 | 0.0 | 0.0 | 0.0 | 0.0 | 0.0 | 0.0 | 0.0 |
| Class 1 | 1.6 | 3.7 | 0.3 | 0.3 | 0.2 | 0.0 | 0.3 | 1.8 |
| Class 2 | 0.0 | 0.0 | 0.0 | 0.0 | 0.0 | 0.0 | 0.0 | 0.0 |
| Class 3 | 1.9 | 4.4 | 0.2 | 0.0 | 0.1 | 0.0 | 0.1 | 1.7 |
| Class 4 | 0.2 | 0.3 | 0.1 | 0.1 | 0.0 | 0.0 | 0.1 | 0.3 |

oretically implies that the features are independent. Under this assumption, the SHAP values estimated by methods such as KernelSHAP (Lundberg & Lee, 2017) must coincide with the analytical Shapley values (5) derived from functional ANOVA (3). We empirically verify this equivalence and this experiment acts as a sanity check.

For the predictive tasks, we trained a Random Forest classifier on the CAR EVALUATION dataset and a MLP on the NURSERY dataset. We compare the exact Shapley values computed via our framework against the approximations produced by KernelSHAP (configured with 200 background samples) and we compute the Integrated Squared Error ($ISE$) between these two indicators over the entire dataset across all classes. For the $i^{\text{th}}$ output class and the $j^{\text{th}}$ feature, the $ISE$ matrix is defined as follows

$$ISE(i, j) := \mathbb{E}\left[\left(\text{KS}_j^{(i)}(\mathbf{X}) - \text{AS}_j^{(i)}(\mathbf{X})\right)^2\right], \quad (21)$$

where $\text{KS}_j^{(i)}$ (resp $\text{AS}_j^{(i)}$) denotes the $j^{\text{th}}$ KernelSHAP value (resp ANOVA-based Shapley value) for the $i^{\text{th}}$ output class. Computing the full decomposition and Shapley values with our closed-form formula takes 0.5 s on CAR EVALUATION and 54 s on NURSERY.

Tables 3 and 4 report the squared errors for each feature and demonstrate, as expected, that in the independent setting, our framework recovers the interpretation given by SHAP in a very competitive time.

**Ground truth study.** We study the MUSHROOMS dataset as a benchmark to validate our approach with a known ground truth, specifically noting the decisive role of features such as *odor*. This dataset represents a typical high-dimensional sparse categorical problem: it contains 22 vari-

ables leading to a hypergrid of $|E| \approx 10^{14}$ configurations, yet only 8 124 samples are observed. Our framework confirms the underlying simplicity of this structure. Indeed, the functional ANOVA decomposition of the probability to classify 1 reveals that the main effects are sufficient to fully reconstruct the signal, achieving an $R^2 \approx 1$ with a negligible MSE of $10^{-15}$ for 0.3 s computation time. Using previous notations, we are in the typical case where $r_{\text{low}} \ll r \ll |E|$, where $r_{\text{low}}$ is the low-rank approximation equal here to 86 and $r$ is the cardinality of the support $\mathcal{X}$ equal to 8 124. Finally, we compute the global feature importance in $\ell_1$ norm (see Fig. 2) by integrating the main effects for each feature $i \in [d]$:

$$\|f_i\|_1 := \sum_{\mathbf{x} \in \mathcal{X}} |f_i(\mathbf{x}_i)| \cdot p_i(\mathbf{x}_i). \quad (22)$$

We compare our method with Observational TreeSHAP (Lundberg et al., 2018). Both methods recover that *odor* (Feature 5), *Gill Color* (Feature 9) and *Spore Print Color* (Feature 20) are the most important features in this dataset. The main differences arise because TreeSHAP exploits in-

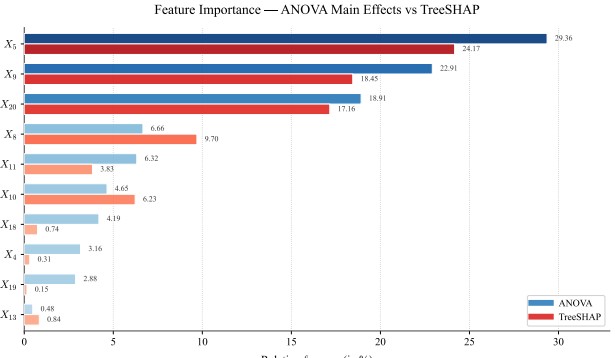

*Figure 2.* Global Feature Importance on MUSHROOMS dataset.

ternal tree paths, while our method only exploits the distribution of the data in a black-box setting.

**High dimensional datasets.** We evaluate our framework on three datasets characterized by high dimensionality and significant sparsity, as detailed in Table 5. In these regimes, the input hypergrid $E$ suffers from a combinatorial explosion, rendering the number of observed samples negligible compared to the theoretical volume ($r \ll |E|$). Consequently, the **full support assumption** of the joint density is violated, preventing the direct application of methods like TreeHFD. Furthermore, high feature correlations in these datasets pose significant challenges for standard Shapley-value based estimators.

*Table 5.* Characteristics of large-scale sparse datasets.

| DATASET | $d$ | $r$ | $|E|$ |
|---|---|---|---|
| POKER HAND | 10 | 25 000 | $52^5$ |
| CONNECT-4 | 42 | 67 000 | $3^{42}$ |
| DOTA2 RESULTS | 113 | 102 000 | $3^{115}$ |

For each task, we trained a specific black-box model to serve as the target function $f$: a Tabular Transformer (POKER), a Random Forest (CONNECT-4), and a very deep fully-connected neural network (DOTA2). We then applied our decomposition framework under the empirical distribution $\widehat{\mathcal{D}}_n$ induced by each dataset.

Table 6 reports the performance in two distinct settings. In the top panel, we restrict the decomposition to the subspace of main effects. We denote by $r_{\mathrm{main}}$ the number of indices in $\mathcal{I}$ to exactly recover all the main effects. Including the intercept, the candidate count is $1+(N_1-1)+\cdots+(N_d-1)$. In the full support case, $r_{\mathrm{main}}$ equals this quantity. In the sparse setting, Algorithm 1 discards dependent functions via the rank check. Our method rapidly isolates these dominant interactions (few seconds), demonstrating efficiency even in high dimensions.

In the bottom panel, we increase the rank approximation to a higher threshold ($r_{\mathrm{high}}$) to evaluate the performance of our method. While increasing the rank improves the explained variance ($R^2$), these results highlight the current computational boundary of our framework. The sheer size of the configuration space in these massive tabular tasks makes high-fidelity reconstruction challenging. The greedy rank selection strategy becomes a bottleneck, suggesting a trade-off between runtime and accuracy when handling vector spaces of this magnitude.

The reported runtime represents a *worst-case* scenario for our current greedy implementation. Despite this, the decomposition remains highly competitive: it processes the full 67 000-sized dataset in under 40 minutes. Furthermore,

*Table 6.* Performance & Metrics. **Top:** Fast approximation restricted to main effects ($r_{\mathrm{main}}$). **Bottom:** Higher rank approximation ($r_{\mathrm{low}}$) for reconstruction analysis.

| DATASET | $r_{\mathrm{main}}$ | Time | $R^2$ | MSE | Rel. MSE |
|---|---|---|---|---|---|
| POKER HAND | 76 | 1.8s | 0.003 | 0.24 | 49% |
| CONNECT-4 | 83 | 10s | 0.45 | 0.07 | 13% |
| DOTA2 | 112 | 23s | 0.36 | 0.02 | 6% |

| DATASET | $r_{\mathrm{high}}$ | Time | $R^2$ | MSE | Rel. MSE |
|---|---|---|---|---|---|
| POKER HAND | 5 000 | 10min | 0.79 | 0.05 | 10% |
| CONNECT-4 | 5 000 | 37min | 0.70 | 0.04 | 7% |
| DOTA2 | 4 000 | 39min | 0.41 | 0.01 | 5% |

when leveraging the inherent spatial structure of the data to optimize the interaction search space, our framework achieves record-breaking efficiency. As illustrated on the experiment below, this optimization allows us to compute a good approximation of the decomposition and explain all 60 000 samples in a very competitive time.

**Feature importance on POKER HAND.** Figure 3 compares the global feature importance produced by our framework with that of attention rollout (Abnar & Zuidema, 2020). Because the rank approximation is necessary in this high-dimensional sparse regime, we set $r_{\mathrm{low}} := 5\,000$, which yields the decomposition $f = \sum_A f_A + f_{\mathrm{res}}$, where $\sum_A f_A$ is the explainable part reconstructed from the $5\,000$ selected basis elements and $f_{\mathrm{res}}$ denotes the unexplained residual. The corresponding local Shapley value for the $i^{\mathrm{th}}$ feature is then defined by attributing each component $f_A$ equally among its members and uniformly distributing the residual across all features:

$$\mathrm{shap}_i := \sum_{A\,:\,A\ni i} \frac{f_A}{|A|} + \frac{f_{\mathrm{res}}}{d}. \qquad (23)$$

Global indices are obtained by aggregating these local attributions in $\ell_1$ norm and normalizing the resulting vector, yielding the percentage contributions reported in Figure 3.

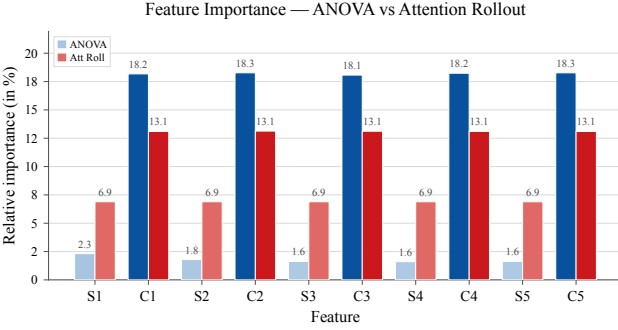

*Figure 3.* Global feature importance on the POKER HAND dataset.

In poker hand classification, the label is determined almost

exclusively by card ranks ($C_i$) rather than suits ($S_i$), a well-established property of the game. Our method recovers this structure sharply, assigning roughly an order of magnitude more importance to ranks than to suits. In contrast, attention rollout distributes attribution far more uniformly across the two groups, substantially overestimating the contribution of suits.

**Binarized MNIST dataset.** We apply our framework to a MLP trained on Binarized MNIST ($60\,000 \times 784$ shaped binary tabular dataset). This high-dimensional setting allows for visual interpretation, focusing on the predicted probability of the digit '3' (see Fig. 1). Table 7 presents our performance results on this big tabular dataset. We achieve high computational efficiency on this dataset by leveraging spatial structure to selectively include relevant subsets. Specifically, we first screen singletons to identify inactive variables (almost surely black pixels) and prune any subset $A$ containing them. Furthermore, we rank active variables by variance to prioritize high-impact coalitions and avoid inefficient searches. Finally, for each variable $i$, we search for interactions within its local spatial neighborhood rather than testing random combinations.

*Table 7.* Performance & Metrics.

|  | $r_{\text{low}}$ | Time | $R^2$ | MSE | Rel. MSE |
|---|---|---|---|---|---|
| Main Effects | 674 | 10s | 0.54 | 0.04 | 41% |
| Low rank | 2 500 | 90s | 0.77 | 0.02 | 20% |
| Middle rank | 5 000 | 300s | 0.83 | 0.01 | 15% |
| High rank | 10 000 | 15min | 0.86 | 0.01 | 12% |

## 6. Discussion

**Conclusion.** We have presented a rigorous framework for computing the exact functional ANOVA decomposition for categorical inputs in a pure *black-box* setting. Our explicit formulation is robust to arbitrary dependence structures and support constraints, preserving the Generalized Functional ANOVA properties. Our approach offers a significant paradigm shift in computational efficiency and incurs a one-time global computation cost. Once the decomposition is obtained, explanations for any number of samples are available instantaneously. Furthermore, this general framework not only encompasses standard results in independent settings but also provides a powerful, theoretically grounded counterpart to approximation-based methods for additive explanation.

**Limitations & Future Work.** Our framework inherits the computational cost intrinsic to ANOVA-style decompositions. In the dense regime, the exact algorithm requires solving a linear system of full hypergrid size, which quickly becomes prohibitive in high dimension; in the sparse regime,

the system size reduces to the observed support, but the greedy rank search then takes over as the dominant bottleneck. Our results on Binarized MNIST demonstrate that this general algorithm can be significantly optimized by leveraging the spatial structure of the data, confirming that incorporating domain knowledge effectively mitigates the curse of dimensionality. Future iterations will focus on formally integrating such structural constraints to further improve computational efficiency, and on extending this exact decomposition framework to continuous domains. A promising algorithmic direction to alleviate the computational complexity is to bypass the greedy selection altogether. One can directly assemble all basis functions up to a prescribed interaction order into a single system and apply sparsity-promoting model selection (e.g., LASSO or LARS) to discard redundant columns. The resulting pruned, full-rank system can then be solved efficiently.

A second limitation, conceptually distinct from the algorithm itself, concerns the non-uniqueness of the decomposition in the sparse regime. Classical uniqueness results for the functional ANOVA decomposition rely on density assumptions (boundedness away from zero) that collapse in the discrete case with finite, possibly sparse, support—a gap the present work begins to fill. In this regime, the overcomplete dictionary is rank-deficient and admits a family of equivalent representations. This non-uniqueness is a geometric property of the data support rather than a methodological artefact: any ANOVA or GAM-like decomposition on sparse categorical inputs is subject to the same identifiability issue. It raises two natural research directions: first, identifying an optimal criterion (e.g., sparsity-promoting or information-theoretic) for selecting a meaningful decomposition among all valid ones; second, developing a principled full-support modeling strategy that restores uniqueness while preserving component reliability.

## Acknowledgements

The authors are grateful for the feedback by the anonymous reviewers, that led to considerable improvement of the paper. This work was partially supported by the French *Association Nationale de la Recherche et de la Technologie* (ANRT) through a CIFRE PhD project at Électricité de France (EDF). Fabrice Gamboa and Jean-Michel Loubes acknowledge support from the ANR-3IA Artificial and Natural Intelligence Toulouse Institute (ANITI).

## Impact Statement

This paper presents work whose goal is to advance the field of trustworthy machine learning. Explicit generalized functional ANOVA enables a theoretical and practical improvement in the post-hoc mechanistic interpretability of complex models with categorical inputs. This yields the ability to produce fine-grained attributions to the information sources used by such a model. This could help improve the selection and refinement of these models, as well as the trust one can place in them.

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

# Appendices

# A. Proofs

### A.1. Proof of Theorem 3.2

The proof of this theorem is long so it will be split into three lemmas. First, we show that the collection $\mathcal{F}$ linearly spans $L^2$ using the two first lemmas. Then, in the last one, we prove that the elements of $\mathcal{F}$ satisfy the hierarchical orthogonality condition (2). Using that $\mathcal{F}$ spans $L^2$, the Fourier expansion (8) becomes direct. Then, with the hierarchical orthogonality condition satisfied by all the $\phi_A^{(\mathbf{z})}$, the corresponding functions $f_A$ satisfy also this condition.

**Lemma A.1.** *For all $i \in [d]$ and $z_i \in \{0, \ldots, N_i - 2\}$ we introduce the function $\psi_i^{(z_i)}$ defined by*

$$\psi_i^{(z_i)} := \mathbf{1}\{\mathbf{X}_i = z_i\} - \mathbf{1}\{\mathbf{X}_i = N_i - 1\}. \tag{24}$$

*By convention, we also introduce $\psi_\emptyset := 1$ almost surely. For all $i \in [d]$, we denote $\mathcal{G}_i$ by*

$$\mathcal{G}_i := \left\{ \psi_\emptyset, \psi_i^{(0)}, \psi_i^{(1)}, \ldots, \psi_i^{(N_i-2)} \right\}. \tag{25}$$

*Note that $\mathcal{G}_i$ always contains exactly $N_i$ elements for $N_i \geq 1$. According to these notations, one has that $\mathcal{G}_i$ span the vector space of function of $\mathbf{X}_i$.*

*Proof.* To prove this result, one just has to show the following identity

$$\forall x_i \in \{0, \ldots, N_i - 1\}, \, \mathbf{1}\{\mathbf{X}_i = x_i\} \in \mathrm{span}\left(\mathcal{G}_i\right) \tag{26}$$

By definition of $\psi_i^{(\cdot)}$, one can sum the equation (24) over all $z \in \{0, \ldots, N_i - 2\}$ and then has

$$\mathbf{1}\{\mathbf{X}_i = N_i - 1\} = \frac{1}{N_i} \cdot \psi_\emptyset - \frac{1}{N_i} \cdot \sum_{z=0}^{N_i-2} \psi_i^{(z)} \tag{27}$$

Then this equation can be used to show that

$$\forall z^\star < N_i - 1, \, \mathbf{1}\{\mathbf{X}_i = z^\star\} = \psi_i^{(z^\star)} + \frac{1}{N_i} \cdot \psi_\emptyset - \frac{1}{N_i} \cdot \sum_{z=0}^{N_i-2} \psi_i^{(z)} \tag{28}$$

Finally, that concludes the proof. $\qquad \square$

**Lemma A.2.** *The collection of functions $\mathcal{F}$ is a family of vectors that linearly span the space $L^2$.*

*Proof.* By standard multilinear algebra, since for each $i \in [d]$ the family $\mathcal{G}_i$ spans the space of real-valued functions of $\mathbf{X}_i$, the collection of pointwise products

$$\{ g_1 \times \cdots \times g_d \mid g_i \in \mathcal{G}_i, \, i \in [d] \}$$

spans the whole space $L^2$. Fix any configuration $(g_1, \ldots, g_d) \in \mathcal{G}_1 \times \cdots \times \mathcal{G}_d$. Let

$$A := \{i \in [d] \, : \, g_i \neq \psi_\emptyset \}.$$

By construction, for each $i \in A$ there exists an index $z_i \in \{0, \ldots, N_i - 2\}$ such that $g_i = \psi_i^{(z_i)}$, while for $i \notin A$ we have $g_i = \psi_\emptyset = 1$. Therefore,

$$g_1 \times \cdots \times g_d = \prod_{i \in A} \psi_i^{(z_i)}.$$

Iterating over all subsets $A \subseteq [d]$ and all index tuples $\mathbf{z} = (z_i)_{i \in A}$ yields

$$L^2 = \mathrm{span}\left\{ \prod_{i \in A} \psi_i^{(z_i)} \, : \, (A, \mathbf{z}) \in \mathcal{I} \right\}. \tag{29}$$

Moreover, for any fixed $A$, each function $\prod_{i \in A} \psi_i^{(z_i)}$ depends only on $\mathbf{X}_A$; hence it belongs to the subspace

$$L_A^2 := \{ g(\mathbf{X}_A) : g \text{ is real-valued} \}.$$

For a fixed $A \subseteq [d]$, define the linear map $M_A : L_A^2 \to L_A^2$ by

$$M_A\big(g(\mathbf{X}_A)\big) := \frac{g(\mathbf{X}_A)}{p_A(\mathbf{X}_A)},$$

where $p_A$ denotes the pmf of $\mathbf{X}_A$. Since $\mathbf{X}_A$ takes values in its own support, we have $p_A(\mathbf{X}_A) > 0$ almost surely. Thus $M_A$ is well-defined as an operator on $L_A^2$. The inverse map is given by multiplication by $p_A(\mathbf{X}_A)$, namely

$$M_A^{-1}\big(g(\mathbf{X}_A)\big) = g(\mathbf{X}_A)\, p_A(\mathbf{X}_A),$$

so $M_A$ is an automorphism of $L_A^2$. Consequently, for each fixed $A$,

$$\text{span}(M_A(\mathcal{G}_A)) = \text{span}(\mathcal{G}_A), \qquad \mathcal{G}_A := \left\{ \prod_{i \in A} \psi_i^{(z_i)} \right\}_{\mathbf{z} \in E_{A^-}}.$$

Applying this equality for every $A$ and combining with (29) yields

$$L^2 = \text{span}\left\{ \underbrace{\frac{1}{p_A} \cdot \prod_{i \in A} \psi_i^{(z_i)}}_{\phi_A^{(\mathbf{z})}} : (A, \mathbf{z}) \in \mathcal{I} \right\},$$

which is the desired conclusion. $\qquad\square$

**Lemma A.3.** *The collection of functions $\mathcal{F}$ satisfies hierarchical orthogonality conditions* (2), *i.e.*

$$\forall A \subseteq [d],\ \forall B \subsetneq A,\ \forall \mathbf{z}^{(A)} \in E_{A^-},\ \forall \mathbf{z}^{(B)} \in E_{B^-}, \quad \left\langle \phi_A^{(\mathbf{z}^{(A)})}, \phi_B^{(\mathbf{z}^{(B)})} \right\rangle = 0 \tag{30}$$

*Proof.* Recall that by definition,

$$\phi_A^{(\mathbf{z})}(\mathbf{X}) = \frac{\prod\limits_{i \in A} \left( \mathbf{1}\{\mathbf{X}_i = z_i\} - \mathbf{1}\{\mathbf{X}_i = N_i - 1\} \right)}{p_A(\mathbf{X}_A)}.$$

This can be written as follows

$$\phi_A^{(\mathbf{z})}(\mathbf{X}) = \frac{1}{p_A(\mathbf{X}_A)} \cdot \prod_{i \in A} \left( (-1)^{\mathbf{1}\{\mathbf{X}_i = N_i - 1\}} \cdot \mathbf{1}\{\mathbf{X}_i \in \{z_i, N_i - 1\}\} \right), \tag{31}$$

and one can notice that

$$\phi_A^{(\mathbf{z})}(\mathbf{X}) \propto \mathbf{1}\{\forall i \in A,\ \mathbf{X}_i \in \{z_i, N_i - 1\}\} \tag{32}$$

Let $B \subsetneq A \subseteq [d]$ and $(\mathbf{u}, \mathbf{v}) \in E_{A^-} \times E_{B^-}$, we will compute $\left\langle \phi_A^{(\mathbf{u})}, \phi_B^{(\mathbf{v})} \right\rangle$ and show that it equals zero.

First, we will compute $\phi_A^{(\mathbf{u})} \cdot \phi_B^{(\mathbf{v})}$ and determine its support denoted $\mathcal{S}(u, v)$. Let be $\mathbf{x} \in \mathcal{X}$, we will denote $\mathbf{x}_A$ its restriction to $A$ and $\mathbf{x}_B$ its restriction to $B$, one has

$$\phi_A^{(\mathbf{u})}(\mathbf{x}) \cdot \phi_B^{(\mathbf{v})}(\mathbf{x}) \propto \mathbf{1}\{\forall i \in A,\ \mathbf{x}_i \in \{\mathbf{u}_i, N_i - 1\}\} \cdot \mathbf{1}\{\forall i \in B,\ \mathbf{x}_i \in \{\mathbf{v}_i, N_i - 1\}\} \tag{33}$$

We introduce the sets $B_{\mathbf{u}=\mathbf{v}}$ and $B_{\mathbf{u}\neq\mathbf{v}}$ defined by

$$B_{\mathbf{u}=\mathbf{v}} := \{i \in B \,:\, \mathbf{u}_i = \mathbf{v}_i\}, \tag{34}$$

$$B_{\mathbf{u}\neq\mathbf{v}} := \{i \in B \,:\, \mathbf{u}_i \neq \mathbf{v}_i\}, \tag{35}$$

and which satisfy $B_{\mathbf{u}=\mathbf{v}} \cup B_{\mathbf{u}\neq\mathbf{v}} = B$. We can notice that since $B \subsetneq A$, the set $A$ decomposes as follows

$$A = B_{\mathbf{u}=\mathbf{v}} \cup B_{\mathbf{u}\neq\mathbf{v}} \cup (A \setminus B). \tag{36}$$

This partition of $A$ allows us to write the following identity

$$\phi_A^{(\mathbf{u})}(\mathbf{x}) \cdot \phi_B^{(\mathbf{v})}(\mathbf{x}) \propto \mathbf{1}\left\{\forall i \in B_{\mathbf{u}=\mathbf{v}} \cup (A \setminus B),\, \mathbf{x_i} \in \{\mathbf{u}_i, N_i - 1\}\right\} \cdot \mathbf{1}\left\{\forall i \in B_{\mathbf{u}\neq\mathbf{v}},\, \mathbf{x_i} \in \{N_i - 1\}\right\}. \tag{37}$$

Finally, the support $\mathcal{S}(\mathbf{u}, \mathbf{v})$ is given by

$$\mathcal{S}(\mathbf{u}, \mathbf{v}) = \{\mathbf{x} \in \mathcal{X} \,:\, \forall i \in B_{\mathbf{u}=\mathbf{v}} \cup (A \setminus B),\, \mathbf{x_i} \in \{\mathbf{u}_i, N_i - 1\} \quad \forall j \in B_{\mathbf{u}\neq\mathbf{v}},\, \mathbf{x_j} \in \{N_j - 1\}\} \tag{38}$$

Given the support, we can compute the scalar product between $\phi_A^{(\mathbf{u})}$ and $\phi_B^{(\mathbf{v})}$ which is basically a sum of $\mathbf{x}$ over this support which components are restricted to $A$ since $\phi_A^{(\mathbf{u})} \cdot \phi_B^{(\mathbf{u})} \in L_A^2$. This simplifies the indicator function and the pmf of $\mathbf{X}_A$, as follows

$$\left\langle \phi_A^{(\mathbf{u})}, \phi_B^{(\mathbf{v})} \right\rangle = \sum_{\mathbf{x}_A \,:\, \mathbf{x} \in \mathcal{S}(\mathbf{u},\mathbf{v})} \frac{\prod\limits_{i \in A} (-1)^{\mathbf{1}\{\mathbf{x}_i = N_i - 1\}} \cdot \prod\limits_{j \in B} (-1)^{\mathbf{1}\{\mathbf{x}_j = N_j - 1\}}}{p_B(\mathbf{x}_B)}. \tag{39}$$

Let $\mathbf{x} \in \mathcal{S}(\mathbf{u}, \mathbf{v})$, the previous product simplifies as follows

$$\prod_{i \in A} (-1)^{\mathbf{1}\{\mathbf{x}_i = N_i - 1\}} \cdot \prod_{j \in B} (-1)^{\mathbf{1}\{\mathbf{x}_j = N_j - 1\}} = \prod_{i \in A \setminus B} (-1)^{\mathbf{1}\{\mathbf{x}_i = N_i - 1\}}. \tag{40}$$

Then, the scalar product is given by

$$\left\langle \phi_A^{(\mathbf{u})}, \phi_B^{(\mathbf{v})} \right\rangle = \sum_{\mathbf{x}_A \,:\, \mathbf{x} \in \mathcal{S}(\mathbf{u},\mathbf{v})} \frac{\prod\limits_{i \in A \setminus B} (-1)^{\mathbf{1}\{\mathbf{x}_i = N_i - 1\}}}{p_B(\mathbf{x}_B)}. \tag{41}$$

To conclude, we will exploit the structure of the support $\mathcal{S}(\mathbf{u}, \mathbf{v})$ and the fact that the denominator $p_B(\mathbf{x}_B)$ only depends on the coordinates indexed by $B$. Combining these two observations, the sum factorizes as

$$\left\langle \phi_A^{(\mathbf{u})}, \phi_B^{(\mathbf{v})} \right\rangle = \sum_{\mathbf{x}_B \,:\, \mathbf{x} \in \mathcal{S}(\mathbf{u},\mathbf{v})} \frac{1}{p_B(\mathbf{x}_B)} \cdot \prod_{i \in A \setminus B} \left( \sum_{\mathbf{x}_i \in \{\mathbf{u}_i, N_i - 1\}} (-1)^{\mathbf{1}\{\mathbf{x}_i = N_i - 1\}} \right). \tag{42}$$

Since $\mathbf{u} \in E_{A^-}$, one has $\mathbf{u}_i \neq N_i - 1$ for every $i \in A$, so each inner sum reduces to

$$\sum_{\mathbf{x}_i \in \{\mathbf{u}_i, N_i - 1\}} (-1)^{\mathbf{1}\{\mathbf{x}_i = N_i - 1\}} = (-1)^0 + (-1)^1 = 0. \tag{43}$$

The strict inclusion $B \subsetneq A$ ensures that $A \setminus B \neq \emptyset$, so the product above contains at least one vanishing factor. Therefore

$$\left\langle \phi_A^{(\mathbf{u})}, \phi_B^{(\mathbf{v})} \right\rangle = 0, \tag{44}$$

which concludes the proof of the hierarchical orthogonality condition. $\square$

## A.2. Proof of Corollary 3.4

The proof of this corollary is based on a linear algebra argument. Indeed, the collection of functions $\mathcal{F}$ can be seen as exactly $|E|$ vectors of $L^2$. Furthermore, in the full support case, $\dim L^2 = |E|$. Finally, since $\mathcal{F}$ spans $L^2$ and has exactly $\dim L^2$ elements, this family becomes a basis.

### A.3. Proof of Corollary 3.5

In full support case, the decomposition is unique, so we can conclude using this simple argument. However, we will still show that in the independent case, $\mathcal{F}$ becomes an orthogonal basis of $L^2$, *i.e.*

$$\forall A \neq B, \forall \left( \mathbf{z}^{(A)}, \mathbf{z}^{(B)} \right) \in E_{A^-} \times E_{B^-}, \left\langle \phi_A^{(\mathbf{z}^{(A)})}, \phi_B^{(\mathbf{z}^{(B)})} \right\rangle = 0 \tag{45}$$

We first have

$$\phi_A^{(\mathbf{z}^{(A)})} \cdot \phi_B^{(\mathbf{z}^{(B)})} = \prod_{i \in A} \phi_i^{\left(\mathbf{z}_i^{(A)}\right)} \prod_{j \in B} \phi_j^{\left(\mathbf{z}_j^{(B)}\right)} \tag{46}$$

We have the following *disjoint* union $A \cup B = (A \cap B) \cup (A \cup B \setminus (A \cap B))$, so we can rewrite the product as follows

$$\phi_A^{(\mathbf{z}^{(A)})} \times \phi_B^{(\mathbf{z}^{(B)})} = \underbrace{\left( \prod_{i \in A \cap B} \phi_i^{\left(\mathbf{z}_i^{(A)}\right)} \times \prod_{i \in A \cap B} \phi_i^{\left(\mathbf{z}_i^{(B)}\right)} \right)}_{\text{function}(\mathbf{X}_{A \cap B})} \times \underbrace{\left( \prod_{i \in (A \cup B) \setminus (A \cap B)} \phi_i^{\left(\mathbf{z}_i^{((A \cup B) \setminus (A \cap B))}\right)} \right)}_{\text{function}(\mathbf{X}_{(A \cup B) \setminus (A \cap B)})} \tag{47}$$

Using the independance and by applying the expectation operator, we have the following *proportional* equality

$$\mathbb{E}\left[ \phi_A^{(\mathbf{z}^{(A)})} \times \phi_B^{(\mathbf{z}^{(B)})} \right] \propto \mathbb{E}\left[ \prod_{i \in (A \cup B) \setminus (A \cap B)} \phi_i^{\left(\mathbf{z}_i^{((A \cup B) \setminus (A \cap B))}\right)} \right] \tag{48}$$

which gives thanks to independence

$$\mathbb{E}\left[ \phi_A^{(\mathbf{z}^{(A)})} \times \phi_B^{(\mathbf{z}^{(B)})} \right] \propto \prod_{i \in (A \cup B) \setminus (A \cap B)} \mathbb{E}\left[ \phi_i^{\left(\mathbf{z}_i^{((A \cup B) \setminus (A \cap B))}\right)} \right] \tag{49}$$

Since all the $\phi_i^{(\cdot)}$ are centered, we have the result.

### A.4. Proof of Remark 3.6

In the uniform boolean case, it is equivalent to suppose that for all $i \subseteq [d]$ the random variables $\mathbf{X}_i$ are independent and follow a $\text{Bernoulli}(1/2)$ distribution. In this very interesting case, it is nice to notice that for all $A \subseteq [d]$ the set $E_{A^-}$ is a singleton, moreover the elements of $\mathcal{F}$ simply become for any $A \subseteq [d]$

$$\phi_A = 2^{|A|} \cdot \prod_{i \in A} \left( \mathbf{1}\left\{ \mathbf{X}_i = 0 \right\} - \mathbf{1}\left\{ \mathbf{X}_i = 1 \right\} \right) = 2^{|A|} \cdot \prod_{i \in A} (-1)^{\mathbf{X}_i} = 2^{|A|} \cdot \chi_A. \tag{50}$$

### A.5. Proof of Proposition 3.10

To prove this result, recall that

$$f(\mathbf{X}) = \sum_{(A, \mathbf{z}^{(A)}) \in \mathcal{I}} c_A^{(\mathbf{z}^{(A)})}(f) \cdot \phi_A^{(\mathbf{z}^{(A)})}(\mathbf{X}).$$

Let $B \subseteq [d]$ and $\mathbf{z}^{(B)} \in E_{B^-}$, we simply compute the scalar product between $f$ and $\phi_B^{(\mathbf{z}^{(B)})}$.

$$\underbrace{\left\langle f, \phi_B^{(\mathbf{z}^{(B)})} \right\rangle}_{\mu_B^{(\mathbf{z}^{(B)})}(f)} = \sum_{(A, \mathbf{z}^{(A)}) \in \mathcal{I}} c_A^{(\mathbf{z}^{(A)})}(f) \cdot \underbrace{\left\langle \phi_A^{(\mathbf{z}^{(A)})}, \phi_B^{(\mathbf{z}^{(B)})} \right\rangle}_{\gamma_{A,B}\left(\mathbf{z}^{(A)}, \mathbf{z}^{(B)}\right)}. \tag{51}$$

Then by stacking by block, we obtain

$$
\underbrace{\boldsymbol{\mu}_B(f)}_{\text{shape } (|E_{B^-}|,1)} = \sum_{A \subseteq [d]} \text{matmul} \left( \underbrace{\Gamma_{B,A}}_{\text{shape } (|E_{B^-}|,|E_{A^-}|)} , \underbrace{\boldsymbol{c}_A(f)}_{\text{shape } (|E_{A^-}|,1)} \right). \tag{52}
$$

This sum can be written as the following matrix multiplication

$$
\boldsymbol{\mu}_B(f) = \text{matmul} \left( \Gamma_{B,[d]}, \boldsymbol{c}_A(f) \right). \tag{53}
$$

Finally, by stacking over all the subsets $B \subseteq [d]$, one has the desired result.

### A.6. Proof of Remark 4.3

This remark applies to the setting where $|\mathcal{X}| = r$ and where a set of indices $\mathcal{S}_r \subseteq \mathcal{I}$ of size $r$ has been selected such that $\mathcal{B}_{\mathcal{S}_r}$ forms a basis of $L^2$, *i.e.*, a spanning family of exactly $r$ elements. In this context, stating that there exists a sub-vector $\mathbf{X}_A$ of $\mathbf{X}$ such that $f(\mathbf{X})$ does not depend on $\mathbf{X}_A$ is equivalent to stating that there exists a function $g$ such that

$$
f(\mathbf{X}) = g(\mathbf{X}_{A^c}), \tag{54}
$$

where $A^c$ denotes the complement of $A$ in $[d]$. We can decompose $f(\mathbf{X})$ in the basis $\mathcal{B}_{\mathcal{S}_r}$ as follows

$$
f(\mathbf{X}) = \sum_{(U,\mathbf{z}) \in \mathcal{S}_r} c_U^{(\mathbf{z})}(f) \cdot \phi_U^{(\mathbf{z})}(\mathbf{X}). \tag{55}
$$

Necessarily, since $f(\mathbf{X})$ is a function solely of $\mathbf{X}_{A^c}$, all Fourier coefficients $c_U^{(\mathbf{z})}(f)$ such that $U \subseteq A$ are zero.

## B. Experiments Details

### B.1. Setup

All experiments were conducted on a MacBook Pro M4 with 32 GB of RAM. The implementation relies on Python, using `NumPy` (Harris et al., 2020) for the computations of $\mathcal{F}$. Specifically, we leverage vectorized linear algebra operations by representing all realizations of $\phi_A^{(\mathbf{z})}$ as a vector of shape $(|\mathcal{X}|, 1)$. The code for reproducing these results is provided in the supplementary material. All tabular datasets studied in this experimental section come from the UCI ML Repository (Kelly et al., 2023) and the MNIST Database (LeCun & Cortes, 1998).

### B.2. Black-Box Setting

For all experiments involving *real-world datasets*—i.e., every experiment excluding the initial synthetic one—we trained a machine learning model to serve as the *black-box* function of interest we want to explain and apply our framework. We detail the architectures and performance metrics for each experiment in the paragraphs below.

**Car Evaluation.** For this dataset, we trained a Random Forest using the `ScikitLearn` (Pedregosa et al., 2011) Python library. All hyperparameters were set to their default values, with the exception of the number of trees, which was set to 100. The model achieved an overall accuracy of 0.9672.

**Nursery.** We trained a Multi-Layer Perceptron (MLP) using the `PyTorch` (Paszke et al., 2019) Python library. Default hyperparameters were used, except for the layer structure, which follows the architecture:

$$
\text{input} \rightarrow 64 \rightarrow 32 \rightarrow \text{output}
$$

The model achieved an overall accuracy of 0.9973.

**Mushrooms.** We trained a tree ensemble using eXtreme Gradient Boosting (XGB, Chen et al. (2015)) via the corresponding Python library. We used exactly 100 trees with a maximum depth of 5, resulting in a perfect overall accuracy of 1.00.

**Poker Hand.** For this dataset, we trained a Tabular Transformer using the `PyTorch` Python library. The model consists of a 3-layer encoder (embedding dimension of 32) followed by a classifier with the following architecture:

$$\text{input} \rightarrow 320 \rightarrow 64 \rightarrow \text{output}$$

The model achieved an overall accuracy of $0.9843$.

**Connect-4.** We trained a Random Forest using the `ScikitLearn` Python library. All parameters were kept at default values, except for the number of trees, which was set to 100. The model achieved an overall accuracy of $0.8192$.

**Dota2 Results.** We trained a very deep fully connected neural network using the `PyTorch` Python library. Default hyperparameters were maintained, except for the layers, which adhere to the following architecture:

$$\text{input} \rightarrow 1024 \rightarrow \cdots \rightarrow 32 \rightarrow \text{output}$$

The model achieved an overall accuracy of $0.5869$.

**MNIST.** For this dataset, we trained a Multi-Layer Perceptron (MLP) using the `PyTorch` Python library. All parameters were set to default values, except for the layers, which follow the architecture:

$$\text{input} \rightarrow 512 \rightarrow 128 \rightarrow \text{output}$$

The model achieved an overall accuracy of $0.9709$. Note that all the images have been binarized before the training after the standard $[0, 1]$ transformation.

## C. Comparison with TreeHFD

We compare our closed-form formulation with the TreeHFD algorithm (Bénard, 2025), which approximates the main and pairwise effects of the functional ANOVA decomposition under the full support assumption (Chastaing et al., 2012). To ensure a fair validation, we adopt a synthetic setting where the theoretical assumptions of TreeHFD are met.

Let $\mathbb{X} \in \mathbb{R}^{n \times d}$ be a dataset drawn from a distribution $\mathcal{D}$, where the dimension $d$ is even. We define the target function $g$ as:

$$g(\mathbf{x}) \coloneqq \left(W_1 \odot \mathbf{x}_{1:d/2}\right)^\top \left(W_2 \odot \mathbf{x}_{d/2:d}\right), \tag{56}$$

where $W_1$ and $W_2$ are fixed random weights. By construction, $g$ contains interactions of at most order 2. Consequently, approximating $g$ is a trivial task for a gradient boosted tree model restricted to depth-2 interactions. We train an XGBoost model (100 trees, max-depth 2) on a synthetic dataset of size $n = 10\,000$. The comparison results are reported in Table 8.

*Table 8.* Comparison between our closed-form ANOVA and TreeHFD.

| $d$ | XGB $R^2$ | Orthogonality | | Differences between indicators | |
|---|---|---|---|---|---|
| | | Closed-Form | TreeHFD | Main Effects | Shapley Values |
| 2 | 0.9975 | $1.54 \times 10^{-19}$ | $1.77 \times 10^{-4}$ | $1.87 \times 10^{-3}$ | $1.33 \times 10^{-5}$ |
| 4 | 0.9966 | $5.65 \times 10^{-18}$ | $4.73 \times 10^{-3}$ | $2.44 \times 10^{-2}$ | $5.57 \times 10^{-4}$ |
| 6 | 0.9966 | $1.42 \times 10^{-16}$ | $8.31 \times 10^{-3}$ | $2.02 \times 10^{-2}$ | $3.81 \times 10^{-4}$ |
| 8 | 0.9899 | $3.57 \times 10^{-16}$ | $8.59 \times 10^{-3}$ | $7.88 \times 10^{-3}$ | $4.89 \times 10^{-5}$ |
| 10 | 0.9790 | $2.72 \times 10^{-16}$ | $6.65 \times 10^{-4}$ | $1.61 \times 10^{-3}$ | $2.26 \times 10^{-6}$ |

To quantify the discrepancy between the two methods, we report two metrics. The Main Effects column measures the Euclidean distance between the vectors of integrated $L_1$ norms of the main effects (in $\mathbb{R}_+^d$) derived from each decomposition. The Shapley Values column reports the integrated squared difference between the Shapley values obtained by both methods across all observations.

While not explicitly reported in Table 8, we verified that both methods achieve perfect reconstruction of the trained XGBoost model. Consequently, the interpretability indices provided by both methods are highly consistent, as shown by the low values

in the difference metrics. We also report the orthogonality metric used by Bénard (2025), defined as the maximum absolute inner product between a component $f_A$ and any of its subsets $f_B$ (where $B \subsetneq A$). Our closed-form solution enforces strict hierarchical orthogonality up to machine precision (errors $\sim 10^{-16}$), whereas TreeHFD, being an approximation, only satisfies this condition with a tolerance of order $10^{-3}$.

