# OpenReview forum: "Exact Functional ANOVA Decomposition for Categorical Inputs Models"
_ICML.cc/2026/Conference — ICML 2026 spotlight_

### Official Review · Reviewer_pgE3 · 2026-03-09

**Soundness:** 4
**Presentation:** 4
**Significance:** 3
**Originality:** 3
**Overall Recommendation:** 5
**Confidence:** 3

**Summary:**

This paper explores the functional ANOVA decomposition of a machine learning model restricted to categorical input features. The authors explicitly construct the decomposition by expanding the Walsh basis to categorical input spaces. The authors then show the "hierarchical orthogonality" condition for this decomposition, and thus provides an explicit construction of the Hooker (2007) functional ANOVA decomposition via projections. This construction can be seen as a direct extension of binary input spaces, where it is well known as the Fourier decomposition of real-valued Boolean functions. The authors then explore computation of this decomposition under sparsity of the input space induced by the observed categorical combinations. The decomposition is showcased on several synthetic and real-world cases, including a case study with known ground truth. Since the decomposition allows to compute concepts such as the Shapley value, the authors empirically compare their approach against approximation methods of the Shapley value. As a generally challenging practical use case, the authors demonstrate their approach on MNIST.

**Compliance With Llm Reviewing Policy:**

Affirmed.

**Final Justification:**

I think this paper provides an interesting and novel insight of explicitly constructing the functional ANOVA using Fourier bases and projections, thereby extending results on several previous works. These decompositions are highly relevant for interpretation, but difficult to compute. The authors also provide an approach to overcome the computational problem, but the main contribution lies in the mathematically rigorous extension of the decomposition to categorical inputs. I recommend acceptance.

**Key Questions For Authors:**

- Could you comment on the novelty of your proposed basis, especially in light of previous work [A]? Is your proposed basis unique with respect to the hierarchical orthogonality condition?

[A] Dutkay, Dorin Ervin, Gabriel Picioroaga, and Sergei Silvestrov. "On generalized Walsh bases." Acta Applicandae Mathematicae 163.1 (2019): 73-90.

**Limitations:**

The authors address some limitations of the method. I think this part could be improved by including future directions with partially continuous input spaces.

**Strengths And Weaknesses:**

**Overall Assessment:** This paper provides a rigorous extension to explicit construction of a functional ANOVA decomposition for categorical input spaces. This is an interesting and novel insight, which is somewhat expected to follow from the binary case, but non-trivial, and rigorously introduced. In practice, this decomposition has two major limitations, since 1) it is severely affected by the curse of dimensionality, and 2) only holds for all-categorical inputs. However, the theoretical advances are non-trivial and the provided case studies are convincing, which is why I recommend acceptance.

**Soundness**: The claims provided in this paper are well-supported by evidence, the theoretical results are rigorously established and nicely demonstrated across different settings. However, it remains unclear, if the provided basis is novel or a unique extension of the Walsh basis to categorical inputs (see question below).

**Presentation**: The presentation of the paper is clear and well written.

**Significance**: This paper provides an important first step towards extensions of explicit functional ANOVA decompositions, clearly discusses its challenges, such as uniqueness. In practice, this approach will mostly work for moderately-sized tabular problems, but provides a fundamental building block to explore more complex extensions for high-dimensional non-tabular settings. Such a first step is clearly illustrated using the MNIST dataset.

**Originality**: The paper bridges the functional ANOVA literature and boolean function literature by expanding the Walsh basis to explicitly construct the functional ANOVA decomposition. It provides a variety of interesting insights.

---

> ### Author Rebuttal · Authors · 2026-03-30
>
> We sincerely thank the reviewer for the positive assessment of our work and for pointing out the relevant reference [A].
>
> *[A] Dutkay, Dorin Ervin, Gabriel Picioroaga, and Sergei Silvestrov. "On generalized Walsh bases." Acta Applicandae Mathematicae 163.1 (2019): 73-90*
>
> ## Question: Novelty of the proposed basis and relation to [A]
>
> Our proposed family is indeed novel, particularly when contrasted with [A]. The construction in Dutkay et al. (2019) provides orthonormal generalized Walsh bases for the continuous space $L^2$, derived from $N$-adic refinements and Cuntz-algebra representations. In contrast, our framework operates on the finite discrete space $E := \prod_{i=1}^d \lbrace 0,\dots,N_i-1\rbrace$ equipped with an arbitrary probability measure, which may have non-rectangular support.
>
> We adopted the _Walsh–Hadamard_ terminology to reflect the strong connection to the well-known Walsh–Hadamard basis (O'Donnell, 2014), defined on the Boolean hypercube by:
> $$\forall \mathbf{x} \in \lbrace 0,1\rbrace ^d,\quad \chi_A(\mathbf{x}) := \prod_{i \in A}(-1)^{\mathbf{x}_i}.$$
> Our family extends this construction to handle arbitrary categorical features with heterogeneous cardinalities and dependent distributions. It exactly recovers the standard Walsh–Hadamard basis if and only if all features are i.i.d. $\mathrm{Bernoulli}(1/2)$.
>
> **Uniqueness with respect to hierarchical orthogonality.** Strictly speaking, the basis itself is not unique: many valid bases can span the functional space while satisfying the hierarchical orthogonality condition (2). However, when the support of $\mathbf{X}$ is full, the ANOVA decomposition itself is unique, and projecting onto any of these valid bases yields the same representation. Conversely, when the support is sparse, the representation is no longer unique, as discussed in Section 4.
>
> **Progressive recovery of classical results.** A key feature of our construction is that it satisfies increasingly strong properties as assumptions are progressively strengthened, naturally recovering all classical results as special cases:
>
> - **a) Full support:** the family becomes a basis and the ANOVA representation is unique (Corollary 3.4).
> - **b) Independence:** the family becomes an orthogonal basis, and the decomposition strictly recovers the standard functional ANOVA given by the Möbius transform of conditional expectations (Corollary 3.5).
> - **c) i.i.d. Bernoulli(1/2):** the family collapses to the standard Walsh–Hadamard parity functions $\lbrace \chi_A \rbrace _{A \subseteq [d]}$, recovering the classical Fourier analysis of pseudo-Boolean functions (Remark 3.6).
>
> In this sense, our family generalizes orthogonal Boolean Fourier analysis to the broadest categorical setting currently tractable. Prior discrete Fourier construction (O'Donnell, 2014) is in fact restricted to case c). Prior works on generalized Hoeffding/ANOVA decomposition (Hooker, 2007; Chastaing et al., 2012; Il Idrissi et al., 2025) rigorously established the existence and uniqueness of this decomposition under specific assumptions but provided no explicit decomposition basis in general.

---

> > ### Author Rebuttal · Reviewer_pgE3 · 2026-04-01
> >
> > I thank the authors for their response and clarifications. I will keep my score.

---

> > > ### Author Response · Authors · 2026-04-07
> > >
> > > Dear Reviewer,\
> > > We thank you for keeping your positive score.\
> > > Thank you again for your time and constructive feedback.\
> > > Best regards,\
> > > Authors

---

### Official Review · Reviewer_ATMn · 2026-03-11

**Soundness:** 4
**Presentation:** 3
**Significance:** 4
**Originality:** 4
**Overall Recommendation:** 6
**Confidence:** 4

**Summary:**

This paper derives an exact functional ANOVA decomposition for models with categorical inputs, including settings with dependent variables and sparse supports. The work proposes a closed-form formulation based on a generalized discrete Fourier representation that allows decomposition of a model prediction into main effects and higher-order interactions without relying on sampling approximations.
While traditional ANOVA decompositions have explicit formulas only under independence assumptions, and dependent inputs typically require expensive sampling approximations, this paper addresses this limitation by constructing a basis for categorical input spaces that satisfies the hierarchical orthogonality conditions of generalized functional ANOVA.

**Compliance With Llm Reviewing Policy:**

Affirmed.

**Key Questions For Authors:**

1 When the support is sparse, the decomposition basis is not unique. How sensitive are the explanations (e.g., feature importance or interaction terms) to the chosen basis?
2 The method becomes computationally expensive for large feature spaces. Are there possible strategies such as randomized linear solvers or regularization of the Gram system to reduce the cost?
3. The framework uses the empirical distribution of inputs. How robust is the decomposition if the empirical distribution poorly represents the true dependency structure?

**Limitations:**

Yes

**Strengths And Weaknesses:**

Strength: This paper had a clear theoretical contribution: a closed-form ANOVA decomposition for categorical inputs with arbitrary dependence, which is a nontrivial extension of classical results that usually require independent inputs. The use of a generalized Fourier basis is mathematically interesting and well motivated. The proposed framework can handle arbitrary dependence structures in categorical features, which extends the traditional ANOVA work significantly and is valuable for tabular datasets. The authors provide a strong mathematical foundation with hierarchical orthogonality conditions, existence and uniqueness results under full support, and explicit basis construction. Experiments on datasets like MUSHROOMS and CONNECT-4 show that the method can recover meaningful feature importance and achieve good reconstruction accuracy under low-rank approximations.

 Weaknesses
1 The method requires solving a linear system of size proportional to the support size or the hypergrid size. Even with the greedy rank-based algorithm, the approach still struggles with very high-dimensional problems. The paper acknowledges that the greedy rank search becomes a computational bottleneck.
2 The decomposition is computed with respect to the empirical distribution of the dataset, which may differ significantly from the true data-generating distribution. This could affect the interpretability of the resulting explanations.
3 When the support is sparse, the basis support is not unique due to feature correlations. As the paper notes, different bases can produce different interpretations even though they yield the same predictions. This raises concerns about the stability and interpretability of the decomposition.
4 The experimental section focuses mostly on reconstruction metrics (R², MSE) which are limited.

---

> ### Author Rebuttal · Authors · 2026-03-30
>
> We sincerely thank the reviewer for the very positive assessment of our work and for raising important questions about the proposed decomposition.
>
> ## Question 1
>
> **Context in the literature.** Uniqueness of the Hoeffding/ANOVA decomposition is typically guaranteed by assumptions on the underlying probability density function. In the continuous setting, classical results require the density to be bounded away from zero (Stone, 1994) or impose regularity conditions (Chastaing et al., 2012; Il Idrissi et al., 2025). While reasonable for continuous distributions, these assumptions collapse entirely in the discrete case with finite and potentially sparse support. To the best of our knowledge, few works have addressed the discrete setting directly, a gap which our paper fills.
>
> **The non-uniqueness is geometric, not algorithmic.** In the sparse regime ($r \ll |E|$), the non-uniqueness is an intrinsic consequence of rank deficiency: the overcomplete dictionary contains more elements than $\dim L^2 = r$. This is dictated by the geometry of the data support, not by a specific methodological choice. Any method attempting an ANOVA/GAM-like decomposition on a sparse categorical support would face the same identifiability issue.
>
> **Open questions.** This non-uniqueness raises two research directions we are actively pursuing:
>
> **(A)** identifying an optimal criterion (e.g., sparsity-promoting or information-theoretic) for selecting a meaningful decomposition among all valid ones,
> **(B)** developing a principled full-support modeling strategy that restores uniqueness while preserving component reliability.
>
> **Empirical stability.** In practice, two cases emerge. First, strong pairwise correlations (e.g., $\mathbf X_1 = \mathbf X_2$ a.s.) create genuine attribution ambiguities that require either an arbitrary choice or domain expertise -- an unavoidable challenge for any explanation method. Second, in the more standard case, sparsity-induced linear dependencies affect individual basis functions but not the aggregated components $f_A$. We do not claim general stability w.r.t. basis choice; this would require a dedicated sensitivity study beyond the current scope. With our canonical ordering, recovered explanations are meaningful and align with domain knowledge on all datasets considered, but we agree that systematic stability analysis is an important direction for future work.
>
> ## Question 2
>
> Three main computational bottlenecks arise: (1) the greedy rank-checking procedure in Algorithm 1; (2) storage of the Gram matrix $\Gamma$, of size $|E| \times |E|$; and (3) inversion of the resulting linear system.
>
> A promising avenue would address all three simultaneously by bypassing the greedy selection altogether: one could directly stack all basis functions up to a given interaction order into a single, potentially rank-deficient system. This eliminates (1) and reduces the system dimension, alleviating (2). LASSO or LARS-based model selection could then discard redundant columns, further reducing the effective matrix size. The pruned full-rank system could then be solved via Conjugate Gradient or SVD, addressing (3). Combined with domain-driven pruning (variance screening, spatial locality constraints), this is a concrete and promising direction for future work.
>
> ## Question 3
>
> We agree that this is a genuine limitation at the population level. However, the black-box model $f$ was trained on the observed dataset and has only ever been exposed to the empirical distribution. The relevant object to explain is therefore the behavior of $f$ on the support it was trained on, not its hypothetical behavior on unseen configurations. In this sense, our decomposition is faithful by construction.
>
> **The discrete setting is fundamentally different from the continuous one.** In the continuous case, one can regularize the density to ensure full suppert, effectively "filling in" unobserved regions by continuity. In the categorical setting, no such smoothing is available: an absent configuration genuinely carries zero empirical probability. If the empirical distribution is a poor proxy for the true dependency structure, then the model itself was trained under this same imperfect proxy. Our framework tends to produce explanations consistent with the model's actual operating regime, which we believe makes for a faithful and actionable form of interpretability.

---

> > ### Author Rebuttal · Reviewer_ATMn · 2026-04-03
> >
> > The theoretical contribution is now better justified. The non-uniqueness issue is properly framed as fundamental rather than methodological. The empirical-distribution argument is convincing and strengthens the interpretability claim. The main remaining weakness is the lack of stability analysis and scalability validation, but the authors acknowledge both and provide reasonable future directions.

---

> > > ### Author Response · Authors · 2026-04-07
> > >
> > > Dear Reviewer,\
> > > We thank you for keeping your positive score.\
> > > Thank you again for your time and constructive feedback.\
> > > Best regards,\
> > > Authors

---

### Official Review · Reviewer_QLWo · 2026-03-12

**Soundness:** 3
**Presentation:** 3
**Significance:** 3
**Originality:** 3
**Overall Recommendation:** 5
**Confidence:** 3

**Summary:**

The paper proposes an explicit, closed-form formulation of the generalized functional ANOVA decomposition for models with categorical inputs under arbitrary dependence between variables. Functional ANOVA decomposes a function into additive contributions corresponding to main effects and higher-order interactions, and it is closely related to additive explanation methods such as SHAP values. While the decomposition is well understood for independent inputs (where a closed-form solution exists), computing it in the general dependent setting typically requires expensive sampling-based approximations.

The authors introduce a closed-form representation based on a generalized Fourier-like expansion for categorical domains. They define a family of basis functions extending the Walsh–Hadamard / parity functions used in Boolean Fourier analysis, and show that this family spans the function space L2 over the categorical support. The resulting expansion yields ANOVA components satisfying the hierarchical orthogonality conditions of generalized functional ANOVA. The coefficients are obtained by solving a linear system derived from inner products with the proposed basis functions.

The framework recovers classical results as special cases, including standard ANOVA for independent variables and the Boolean Fourier decomposition under uniform binary inputs. The authors also discuss practical computation in high-dimensional settings with sparse empirical support and propose a greedy rank-based algorithm to construct a basis when the support is smaller than the full hypergrid.

Experiments include synthetic verification, comparisons with KernelSHAP in independent settings, and applications to several categorical datasets and binarized MNIST. Results suggest that the proposed method can efficiently recover additive explanations and main effects even in high-dimensional sparse regimes.

**Compliance With Llm Reviewing Policy:**

Affirmed.

**Final Justification:**

Given that the authors incorporate the clarifications and additions mentioned in their rebuttal into the final manuscript, I consider my concerns to be addressed and am willing to raise my overall score accordingly.

**Key Questions For Authors:**

- Clarification of novelty. How does the proposed basis construction fundamentally differ from existing orthogonal expansions used in generalized Hoeffding decompositions or discrete Fourier analysis on categorical domains? A clearer comparison with prior theoretical constructions would help establish novelty.
- Rank and basis construction. Could the authors clarify more explicitly the notion of rank in Algorithm 1? Does it relate to the dimension of the function space over the empirical support, and if so how?
- Impact of non-unique decompositions in sparse-support settings. When r << |E|, multiple bases can produce valid decompositions. How sensitive are the resulting explanations to the particular basis chosen by the greedy algorithm? Have the authors evaluated the stability of explanations under different basis selections?
- Scalability and complexity. What is the computational complexity of the greedy basis selection procedure in terms of d, r and |E|? Are there any theoretical guarantees or heuristics ensuring scalability in very high-dimensional categorical problems?
- Comparison with existing explanation methods. The appendix includes a comparison with TreeHFD and the main text compares against KernelSHAP in independent settings. Could the authors further clarify how the proposed method compares empirically to other explanation approaches for tabular models, particularly in the presence of feature dependencies? It would be helpful to understand the practical advantages of the proposed approach relative to existing methods for computing interaction effects or feature attributions.
- Clarification on rmain: The paper introduces rmain, the number of basis functions required to exactly recover all main effects. Could the authors clarify how this quantity is computed in terms of the categorical cardinalities of the variables? In particular, since each single-variable component requires multiple basis functions, it would be helpful to explicitly describe how the enumeration in Algorithm 1 ensures that all main-effect components are recovered when using rmain.
- Dependence on reference category choice. The basis construction in Definition 3.1 relies on selecting reference values for each categorical variable to define the sets EA-. While the span of the resulting basis is unaffected by this choice, it is less clear whether the intermediate ANOVA components obtained by grouping basis functions are invariant to the particular reference categories selected. Could the authors clarify whether the resulting decomposition (and the derived interaction effects or Shapley values) is independent of this choice, or whether different reference encodings may lead to different decompositions on finite empirical supports?

**Limitations:**

Yes

**Strengths And Weaknesses:**

Strengths

- Theoretical contribution. The paper proposes a novel formulation of the generalized functional ANOVA decomposition for categorical inputs. The use of a generalized Fourier-type basis extending Walsh–Hadamard parity functions is an interesting idea. The framework unifies several known special cases (standard ANOVA, Boolean Fourier analysis, SHAP under independence), which provides useful conceptual clarity.
- Explicit construction under dependencies. A key contribution is the derivation of a closed-form representation that applies to arbitrary dependence structures between categorical variables. Prior approaches often rely on approximations or Monte Carlo sampling in these cases, so an explicit formulation is potentially valuable.
- Connection to explainability methods. The paper leverages the well-known relationship between functional ANOVA decompositions, Harsanyi dividends, and Shapley values. While this connection itself is not new, the proposed decomposition provides a mechanism that could enable computing these quantities exactly in categorical domains.
Handling sparse supports. The discussion of the sparse-support setting (where the empirical support is much smaller than the full combinatorial grid) is practically relevant for tabular datasets. The idea of selecting a basis from an overcomplete dictionary is intuitive and aligns with the structure of real-world categorical data.
- Empirical demonstrations across settings. The experiments cover several types of datasets, including synthetic setups, standard tabular benchmarks, and high-dimensional examples (e.g., binarized MNIST). These examples help illustrate how the method behaves under different dimensionalities and sparsity regimes.

Weaknesses

- The relationship to existing orthogonal basis constructions could be clarified further. While the authors position their work as a closed-form solution to generalized ANOVA for categorical inputs, it is not entirely clear how fundamentally new the construction is relative to prior work on Hoeffding decompositions for dependent variables or Fourier expansions on discrete spaces.
- Ambiguity around the notion of rank and Algorithm 1. Algorithm 1 selects basis functions by greedily increasing the rank of the evaluated vectors over the empirical support. While this approach is reasonable, the role of “rank” and its relation to the dimension of the function space could be explained more clearly. The algorithm effectively constructs a linearly independent set of basis vectors over the observed support, but this interpretation is not stated explicitly.
- Ambiguity around identifiability and basis selection. When the support is sparse (r≪∣E∣), the decomposition is not unique. The paper acknowledges this and resolves it with a greedy basis selection procedure. However, the implications for interpretability are not fully discussed: different bases may yield different interaction decompositions, which could affect explanations.
- KernelSHAP comparison provides limited insight. The comparison with KernelSHAP in the independent setting mainly shows that KernelSHAP approximates the exact Shapley values computed by the proposed method. However, this experiment does not strongly demonstrate advantages of the proposed approach beyond computing exact quantities when the decomposition is tractable.
- Computational scalability concerns. Although the authors discuss sparse-support regimes and low-rank approximations, the approach still relies on constructing vectors and computing ranks in spaces whose size scales with the hypergrid. The greedy rank-based algorithm may become computationally expensive in very high-dimensional settings, and the paper does not provide complexity analysis or comparisons with alternative explanation methods in these regimes.
- Experimental evaluation could be stronger. The empirical results mainly demonstrate feasibility rather than clear advantages over existing approaches. In particular:
  - KernelSHAP comparisons have the aforementioned limitations.
  - There is little evaluation of explanation quality or usefulness beyond reconstruction metrics.
  - Runtime comparisons with other explanation techniques are limited.
- Presentation issues. Some parts of the paper are difficult to follow due to dense notation and limited intuition. For example, the construction and interpretation of the basis functions A(z) could benefit from additional explanation or examples. Similarly, the transition between the Fourier expansion and the ANOVA decomposition is not always clearly motivated.

---

> ### Author Rebuttal · Authors · 2026-03-30
>
> We sincerely thank the reviewer for the positive assessment of our work and for recognizing the strengths of our work. The breadth of questions raised reflects genuine engagement with our contribution, which we appreciate. We hope the responses below address all concerns.
>
> ## Q1.
> We provide a family of functions adapted for categorical data under arbitrary distributions $p$, transforming a nonparametric problem into a linear one. It satisfies increasingly strong properties as assumptions are strengthened: **a)** full support $\Rightarrow$ basis, unique ANOVA; **b)** independence $\Rightarrow$ orthogonal basis, standard functional ANOVA; **c)** i.i.d. Bernoulli(1/2) $\Rightarrow$ Walsh–Hadamard parity functions, classical Boolean Fourier analysis (O'Donnell, 2014). Prior discrete Fourier constructions are restricted to c). Prior generalized ANOVA works (Hooker, 2007; Chastaing et al., 2012; Il Idrissi et al., 2025) established existence and uniqueness but provided no explicit basis in general.
>
> ## Q2.
> Let $r$ denote the number of distinct observed configurations, which is also $\dim L^2$. Evaluating the full family on these $r$ points yields a matrix of size $r \times |E|$ with rank $r$. Algorithm 1 greedily selects $r$ linearly independent columns. The rank check at line 10 includes a candidate iff its evaluation vector is linearly independent from previously selected columns. We will clarify this in the manuscript.
>
> ## Q3.
> The non-uniqueness is geometric, not algorithmic: when $r \ll |E|$, the dictionary is overcomplete. Any exact ANOVA-like method on a sparse categorical support faces the same identifiability issue. Empirically, under strong correlations (e.g., $\mathbf{X}_1 = \mathbf{X}_2$ a.s.), attribution ambiguities are unavoidable; in the standard case, sparsity-induced dependencies affect individual basis functions but not the aggregated components $f_A$. We do not claim general stability: a dedicated sensitivity study is an important future direction, alongside two open questions:
>
> **(A)** optimal selection criteria (sparsity-promoting or information-theoretic),
> **(B)** principled full-support strategies restoring uniqueness.
>
> ## Q4.
> Worst-case complexity: $O(|E| * r^2)$: $|E|$ candidates with Gram–Schmidt in $O(r^2)$; dependence on $d$ is hidden in $|E| = \prod_{i=1}^d N_i$. Tractability is achieved via (i) low-rank approximation ($r_{\mathrm{low}} \ll r$), and (ii) the sparsity-of-effects principle, prioritizing low-order interactions in the canonical ordering.
>
> ## Q5.
> We provide two new comparisons under feature dependencies.
>
> *ANOVA vs. Attention Rollout (Poker Hand).* In this task, the label is primarily determined by card ranks ($C_i$) rather than suits ($S_i$), which is domain knowledge.
>
> |       | $S_1$ | $C_1$ | $S_2$ | $C_2$ | $S_3$ | $C_3$ | $S_4$ | $C_4$ | $S_5$ | $C_5$ |
> |-------|------:|------:|------:|------:|------:|------:|------:|------:|------:|------:|
> | ANOVA |  4.50 | 17.38 |  2.51 | 17.43 |  2.05 | 17.34 |  1.97 | 17.42 |  1.95 | 17.45 |
> | Attn  |  6.56 | 13.44 |  6.56 | 13.42 |  6.56 | 13.43 |  6.56 | 13.45 |  6.56 | 13.45 |
>
> Our decomposition sharply separates ranks from suits; Attention Rollout yields a flatter contrast as it does not disentangle functional contributions from internal routing.
>
> *ANOVA vs. Observational TreeSHAP (Mushroom).* Top-10 features (% importance):
>
> |          | $X_5$ | $X_9$ | $X_{20}$ | $X_8$ | $X_{11}$ | $X_{10}$ | $X_{18}$ | $X_4$ | $X_{19}$ | $X_{13}$ |
> |----------|------:|------:|---------:|------:|---------:|---------:|---------:|------:|---------:|---------:|
> | ANOVA    | 29.36 | 22.91 |    18.91 |  6.66 |     6.32 |     4.65 |     4.19 |  3.16 |     2.88 |     0.48 |
> | TreeSHAP | 24.17 | 18.45 |    17.16 |  9.70 |     3.83 |     6.23 |     0.74 |  0.31 |     0.15 |     0.84 |
>
> Both methods agree on the top-3 (odor, gill color, spore print color). Differences arise because TreeSHAP exploits internal tree paths, while our method is black-box. Key advantages of our framework: (i) amortized cost: once computed, all explanations are instantaneous; (ii) model-agnostic.
>
> ## Q6.
> Including the intercept, the candidate count is $1 + \sum_{i=1}^d (N_i - 1)$. In full support, $r_{\mathrm{main}}$ equals this quantity. In the sparse setting, Algorithm 1 discards dependent functions via the rank check. Example: on Mushroom, $1 + 95 = 96$ candidates yield $r_{\mathrm{main}} = 86$. We will add this formula to the manuscript.
>
> ## Q7.
> The "last category" is purely conventional. In **full support**, different reference categories yield collinear bases producing identical ANOVA components and Shapley values. In **sparse support**, different encodings may lead to different decompositions, but this stems from intrinsic rank deficiency, not the reference convention. Experimental consistency across all datasets (e.g., odor on Mushroom, rank-vs-suit on Poker) confirms that recovered importances reflect genuine model structure.

---

> > ### Author Rebuttal · Reviewer_QLWo · 2026-04-04
> >
> > Thank you to the authors for their responses. I believe they have addressed my main concerns adequately.
> >
> > The clarification of the notion of rank and the interpretation of Algorithm 1 seems satisfactory to me. The explanation of how the greedy selection operates and how quantities such as $r$ and $r_{main}$ arise is also clearer.
> >
> > The discussion of non-uniqueness in sparse-support settings is also satisfactory. I appreciate the clarification that this issue is intrinsic to the geometry of the support rather than a consequence of the method. The acknowledgment that stability with respect to basis selection is not guaranteed, together with the identification of this as an important direction for future work, is appropriate.
> >
> > Regarding scalability, the authors provide a clearer characterization of the computational aspects and the role of low-rank approximations. While computational cost remains a limitation, this is now more transparently discussed.
> >
> > Finally, the additional empirical comparisons under feature dependencies (with Attention Rollout and TreeSHAP) are helpful and strengthen the experimental section by better situating the method relative to existing approaches.
> >
> > **Follow-up question:**
> >
> > Regarding the KernelSHAP comparison, there is one point that was originally raised as part of my weaknesses (although I did not explicitly include it in the questions). This concern is mainly about its role in the paper rather than anything else. As presented, it seems to simply show agreement between KernelSHAP and the exact Shapley values computed by the proposed method. Could the authors clarify what they see as the main takeaway of this experiment, and how it supports the overall contribution? This is not a major concern, but I would appreciate the authors’ perspective.
> >
> > Given that the authors incorporate the clarifications and additions mentioned in their rebuttal into the final manuscript, I consider my concerns to be addressed and am willing to raise my overall score accordingly.

---

> > > ### Author Response · Authors · 2026-04-04
> > >
> > > Dear Reviewer,\
> > > We thank you for raising your score to _Accept_ and for this clarification request.
> > >
> > > The main takeaway of this experiment is indeed a **sanity check**, and we agree this deserves to be made more explicit. The logic is the following:
> > >
> > > - In the independent setting, the classical connection between SHAP and ANOVA-based Shapley values guarantees that both quantities coincide theoretically.
> > > - Corollary 3.5 states that our closed-form decomposition exactly recovers the orthogonal ANOVA formulation under feature independence, which in turn implies that our ANOVA-based Shapley values (Eq. 5) must coincide with standard SHAP values.
> > > - The experiment therefore verifies, empirically, that this theoretical connection holds: the residual discrepancies reported in Tables 3–4 are consistent with the stochastic approximation error of KernelSHAP (200 background samples), as expected.
> > >
> > > We specifically chose KernelSHAP as the comparison baseline because it is the canonical *model-agnostic* estimator of SHAP values without relying on model-specific shortcuts such as TreeSHAP or DeepSHAP. This makes the comparison fair and meaningful: both methods operate in the same black-box setting and target the same theoretical quantity.
> > >
> > > In other words, the experiment illustrates how our general framework properly reduces to the well-understood independent case before extending it to arbitrary dependence structures.
> > >
> > > We will clarify the framing of this experiment in the revised version accordingly.
> > >
> > > Thank you again for your time and constructive feedback.\
> > > Best regards,\
> > > Authors

---

### Official Review · Reviewer_ctZq · 2026-03-12

**Soundness:** 3
**Presentation:** 3
**Significance:** 3
**Originality:** 3
**Overall Recommendation:** 5
**Confidence:** 4

**Summary:**

This paper considers the problem of computing a generalized fANOVA decomposition for interpreting machine learning models. The generalized fANOVA is applicable to any datasets with arbitrary feature dependence structures, and it can reveal the complete feature effect and interaction structure of a model, therefore such a decomposition reveals a lot of insights about a model, and the generalized fANOVA is in practice considered computationally extremely demanding, if not intractable. This paper introduces the first explicit closed-form algorithm for computing a generalized fANOVA for categorical input features, but which otherwise can be applied to arbitrary datasets.

The main idea of the proposed algorithm is to extend Fourier analysis for Boolean functions to categorical variables with more than 2 categories. The specific algorithm enumerates the total space of functions on the input space, which is finite-dimensional for categorical features, using a basis that respects the hierarchical orthogonality conditions which define the generalized fANOVA decomposition. A representation of the given model with respect to this basis (which the paper computes as a solution to a system of linear equations) then automatically respects these conditions and therefore delivers the desired decomposition. These results (Existing and properties of this basis and of the decomposition) are mathematically proven in the paper, and are also validated in a small simulation study. The paper also proves (both theoretically and empirically) that this method recovers the standard fANOVA decomposition for independent input variables, as well as the (known) close connection of the fANOVA to Shapley values. This way, the proposed algorithm also offers a way to compute Shapley values for non-independent input features.

Next, because the developed algorithm is computationally extremely costly, the paper proposes to greedily approximate the complete decomposition with terms of low interactions order, and shows on four different UCI datasets that this can indeed lead to meaningful and reasonably accurate approximations with reasonable computational effort. Lastly, the paper also briefly introduces a heuristic improvement of the low-rank approximation for structured data and demonstrates a proof-of-concept on the MNIST dataset.

**Compliance With Llm Reviewing Policy:**

Affirmed.

**Final Justification:**

Raised to "accept"  after the last point of concern for me was resolved in the last rebuttal answer by the authors. Done.

**Key Questions For Authors:**

(1) The choice of using the last point as the "excluded point" in Eq. (7) / for the truncated grid is w.l.o.g., but seems arbitrary, although it will make a difference for the calculated result in practice, right? Meaning that in the end, the calculated decomposition is not unique, but depends on which category has been "excluded" for every feature, right?

(2) The decomposition in Eq. (8) / in Thm. 3.2: Can one make this unique / make the basis functions linearly independent by simply restricting to the subset of \phi_A^(z) with z inside the support? Similar in spirit to the low-rank approximation algorithm?

**Limitations:**

The authors address all important limitations in the paper, however, my impression is that the computational cost of this algorithm does still limit its practical application, my impression is that this could have been made clearer. Whether the low-rank approximation is good enough to make the whole method as a whole "efficient" is in my view difficult to tell; I find the part "High-Dimensional Sparse Datasets" in section 5 still a little bit inconclusive in this regard, and a furthergoing benchmark study in future work would help here (although this is surely out of scope of this submission).

**Strengths And Weaknesses:**

Overall, I did enjoy reading the paper, because it showed a very good new idea and constructed a new algorithm for a difficult and very fundamental problem of interpretable machine learning. The experiments also indicate that this new algorithm may in practice actually indeed enable computation of a fANOVA decomposition for tasks with some more complicated dependence structure and bigger datasets than was possible before.

There are minor issues throughout the manuscript, both language-wise and technical issues, both of which should be improved, but do not affect the overall results. More importantly, it may be that some work not discussed is actually relevant for the method, and it still remains a little unclear to me how computationally efficient and suitable for practical application the developed algorithm in the end is. There is clearly a trade-off here, which in my opinion could have been pointed out more clearly.

Soundness:
The theoretical / mathematical claims are all proven in the paper (incl. the appendix), and I have checked that the proofs are correct. Unfortunately, there are some technical errors in the proofs or in the calculations, very few of them significant, and they all don't change any results, but should be corrected for publication.

Specific questions / remarks:

(M1) In the introduction of section 4, d < 15 should correspond to at least |E| < 10^8, or |E| < 10^4 should correspond to roughly d < 8.

(M2) In Tables 3 and 4 in section 5, some features do not have as much classes, so there must be "empty" fields in these tables, but these are not indicated. I.p., on CAR EVALUATION, the last 3 variables each only have 3 classes, how can they have non-zero error on more than 3 classes?

(M3) Also for tables 3 and 4: What is the unit of the loss here? Is this in percentages?

(M4) In the last equation / the last line of the proof of Lemma A.3, I think this has to be written differently. I think, instead, one has to choose one i_1 \in A\B (always exists!), and then split only according to the value of i_1. As it currently stands in the manuscript, the two conditions "\forall i \in A\B: x_i = u_i" and "\forall i \in A\B: x_i = N_i - 1" do not form a partition for the sum.

Concerning the empirical results: These are reasonably extensive, cover different cases, also more realistic ones with bigger datasets. Overall, the experimental setup appears flawless, and the experimental results do substantiate the claims of the paper.

The authors have provided the code for reproducing all their experimental findings, although I have not reproduced the results myself. Otherwise, without code, the experiments would also be reproducible with the information provided in the appendix.

Presentation:
The paper is clearly structured, the reader is guided well through the paper, and the mathematical and technical ideas are very clearly presented. The presentation is overall good and adequate. Unfortunately, there are a few language issues, which can interrupt the reading flow, but these are minor issues.

There seem to be a few more connections to other conceptual ideas or related work in the field which were not discussed, and the paper would probably benefit from discussing them (even though that surely would not fit in the page limit of the current paper). Specifically:

Accumulated Local Effects (ALE), Apley & Zhu, 2020. This method also offers an efficient algorithm for computing a functional decomposition for dependent input variables, a computation time comparison might be an interesting part of future work. Maybe difficult for unordered categorical features, but nevertheless.
Lengerich, e.a.: Purifying Interaction Effects with the Functional ANOVA: An Efficient Algorithm for Recovering Identifiable Additive Models, AISTATS, 2020. Although not strictly necessary, but my impression is this work could also have at least been mentioned briefly.
Significance:
The paper tries to develop a tractable and even efficient algorithm for calculating a generalized fANOVA decomposition and thereby also Shapley values in a very general setting (arbitrary dependence structures on categorical features). Such an algorithm would be a great improvement in the field of model-agnostic machine learning interpretability, therefore the paper addresses a relevant, very interesting and (if solvable) promising problem. Even though the algorithm developed here may face limited applicability (computational cost / approximation not good enough, or restricted to categorical features), the paper already indicates straightforward further work on lifting these limitations. I.p. further work on continuous features may still increase the significance of this work.

One weakness here: As the authors themselves describe, the algorithm is computationally extremely costly (as is usually always the case for algorithms computing a full fANOVA or other functional decompositions). In its vanilla form, a system of linear equations of size |E|, i.e., of the size of the input space, is solved. So the algorithm cannot be directly used to solve the problem in practical applications. However, this is on the one hand an inherent problem to fANOVA-like decompositions, and the approximations developed by the authors are reasonable and effective for the examples they consider and demonstrate.

Originality:
The paper developes a (mostly completely) new algorithm for computing a generalized fANOVA decomposition, and the main idea used is to adapt the theory of Fourier analysis for binary functions to the setting of general categorical variables, which (to my knowledge) was not done before in the context of machine learning. The paper explores this new idea mathematically very well.

Reason for "weak accept":
This submission is could a "5: Accept" once all the small errors, small technical mistakes and language shortcomings have been improved, in particular the ones mentioned under "Soundness".

---

> ### Author Rebuttal · Authors · 2026-03-30
>
> We sincerely thank the reviewer for the thorough review and constructive suggestions. We address each point below and will integrate all corrections in the revised manuscript.
>
> ## W1) Threshold for full system resolution
>
> Indeed, our formulation was unclear. We will revise the statement to specify that the full linear system can be solved directly for up to $|E| \approx 10^4$ on standard hardware.
>
> ## W2) Tables 3 and 4
>
> We acknowledge the lack of clarity. Each row reports the Integrated Squared Error (ISE) between ANOVA-based Shapley values and KernelSHAP for a given **output class function** $f_i(\mathbf{x}) := Pr(y_{\mathrm{pred}} = i \mid \mathbf{x})$ (basically the $i^{th}$ element of the softmax produced by the trained model $f$). We have 4 classes for CAR EVALUATION and 5 for NURSERY. The ISE is thus reported per output class, not per number of categories of the input features. Formally, for class $i$ and feature $j$:
> $$\mathrm{ISE}(i,j) := \mathbb{E} _ {\mathbf X}\left[\left(\mathrm{Sh}^{\mathrm{Kernel}} _ {j}(f_i, \mathbf{X}) - \mathrm{Sh}^{\mathrm{ANOVA}} _ {j} (f_i, \mathbf{X})\right)^2\right], $$
> this is basically the distance over all the configurations between KernelSHAP and the Shapley values derived from our formula for each output functions $f_i$ and each feature $\mathbf x_j$.
> We will add this definition and an explanatory sentence in the revised manuscript.
>
> ## W3) Proof of Lemma A.3
>
> We thank the reviewer for identifying this error. Indeed, the two conditions do not form a valid partition for the sum. Instead of trying to partition the entire subset $A \setminus B$, one can simply choose a single index $i_1 \in A \setminus B$ (which exists since $B\subsetneq A$) and split the sum based on the value of $x_{i_1}$, which leads to:
> $$\left\langle \phi_A^{(\mathbf u)} , \phi_B^{(\mathbf v)} \right\rangle \propto \sum_{x_{i_1} \in \lbrace \mathbf u_{i_1}, N_{i_1} - 1\rbrace} (-1)^{\mathbb{1}\lbrace x_{i_1} = N_{i_1} - 1\rbrace} = (-1)^0 + (-1)^1 = 1 - 1 = 0.$$
>
> ## W4) Language issues
>
> We will carefully proofread the manuscript.
>
> ## W5) Additional references
>
> **ALE (Apley & Zhu, 2020).** ALE provides efficient visualization of feature effects under dependence in the continuous setting, but relies on a local neighborhood ordering unavailable for unordered categorical variables. Moreover, ALE produces marginal effect plots rather than a full functional decomposition. The two frameworks thus address complementary settings; a computational comparison in the ordered/continuous extension of our work is an interesting future direction.
>
> **Purifying Interaction Effects (Lengerich et al., 2020).** This work proposes an iterative purification algorithm to recover identifiable additive components, closely related to TreeHFD (Bénard, 2025). However, it is restricted to piecewise-constant functions (tree-based models), whereas our framework is fully model-agnostic.
>
> We will include both references in the revised manuscript.
>
> ## Q1) Choice of excluded category
>
> The "last category" is purely conventional: since categorical features have no intrinsic ordering, any category can serve as reference. In the **full support** setting, the decomposition is unique regardless of this choice — different encodings yield the same ANOVA components. In the **sparse support** setting, non-uniqueness arises from the rank deficiency of the Gram matrix $\Gamma$, not from the reference convention. Different encodings may lead to different basis selections in Algorithm 1, but this reflects intrinsic collinearity rather than sensitivity to an arbitrary convention. Our experimental results confirm this: recovered feature importances are stable and consistent with domain knowledge across all datasets.
>
> ## Q2) Restricting to $\mathbf z$ inside the support
>
> The lack of uniqueness is driven by the geometry of the restricted support $\mathcal X$, not by the presence of indices $\mathbf z$ outside it. On a sparse support $\mathcal X$, the family $\lbrace \phi_A^{(\mathbf z)}\rbrace _{ A \subseteq [d] }$ may remain overcomplete even if restricted to the $\mathbf{z}$ within $\mathcal{X}$, and distinct atoms can become linearly dependent when viewed as functions on the support. Uniqueness is automatic only in the full-support case, where the family has cardinality $|E| = \dim L^2$. In the general case, an explicit rank-selection step is needed to extract a linearly independent subfamily of cardinality $r$.
>
> Moreover, even if such a restriction produced a linearly independent subfamily, it would represent one valid basis among many, with no a priori reason to prefer it. This connects to two open directions we are investigating:
>
> **(A)** identifying an optimal selection criterion (e.g., sparsity-promoting or information-theoretic) for a meaningful decomposition,
>
> **(B)** developing a principled full-support modeling strategy to restore uniqueness while preserving component reliability.

---

> > ### Author Rebuttal · Reviewer_ctZq · 2026-04-03
> >
> > Dear authors, thanks. You have resolved (nearly) all of my concerns.
> > I would also update the score to "accept" then as indicated.
> >
> > But I would like this not to be misunderstood/mischaracterized later:
> >
> > You said:
> > "Moreover, ALE produces marginal effect plots rather than a full functional decomposition. The two frameworks thus address complementary settings."
> >
> > That's not really true, even though most people use it that way. Please check page 7, "remark 2", also appendix B.
> >
> > https://arxiv.org/pdf/1612.08468

---

> > > ### Author Response · Authors · 2026-04-04
> > >
> > > Dear Reviewer,\
> > > We thank you for your positive reassessment and for raising your score to _Accept_. We also appreciate you pointing us to this important clarification regarding ALE, which we will properly include in the revised version of our manuscript.\
> > > Thank you again for your time and constructive feedback.\
> > > Best regards,\
> > > Authors

---

### Decision · Program_Chairs · 2026-04-30

**Decision:**

Accept (spotlight)

**Comment:**

All reviewers agree that this is a solid paper presenting interesting results on what is a highly relevant topic. I fully agree. I hope and expect the authors will use the reviews, the points raised during the rebuttal discussion, as well as the new results they presented themselves, when finalizing the camera ready copy.

Great work!